# Single-molecule imaging of PI(4,5)P$_2$ and PTEN in vitro reveals a positive feedback mechanism for PTEN membrane binding

Daisuke Yoshioka[1,2,4], Seiya Fukushima [1,2,4], Hiroyasu Koteishi [2], Daichi Okuno[2], Toru Ide[3], Satomi Matsuoka [1,2,4✉] & Masahiro Ueda [1,2,4✉]

PTEN, a 3-phosphatase of phosphoinositide, regulates asymmetric PI(3,4,5)P$_3$ signaling for the anterior-posterior polarization and migration of motile cells. PTEN acts through posterior localization on the plasma membrane, but the mechanism for this accumulation is poorly understood. Here we developed an in vitro single-molecule imaging assay with various lipid compositions and use it to demonstrate that the enzymatic product, PI(4,5)P$_2$, stabilizes PTEN's membrane-binding. The dissociation kinetics and lateral mobility of PTEN depended on the PI(4,5)P$_2$ density on artificial lipid bilayers. The basic residues of PTEN were responsible for electrostatic interactions with anionic PI(4,5)P$_2$ and thus the PI(4,5)P$_2$-dependent stabilization. Single-molecule imaging in living *Dictyostelium* cells revealed that these interactions were indispensable for the stabilization in vivo, which enabled efficient cell migration by accumulating PTEN posteriorly to restrict PI(3,4,5)P$_3$ distribution to the anterior. These results suggest that PI(4,5)P$_2$-mediated positive feedback and PTEN-induced PI(4,5)P$_2$ clustering may be important for anterior-posterior polarization.

[1] Department of Biological Sciences, Graduate School of Science, Osaka University, Toyonaka, Osaka 565-0043, Japan. [2] Center for Biosystems Dynamics Research (BDR), RIKEN, Suita, Osaka 565-0874, Japan. [3] Graduate School of Natural Science and Technology, Okayama University, Okayama-shi, Okayama 700-8530, Japan. [4] Graduate School of Frontier Biosciences, Osaka University, Suita, Osaka 565-0871, Japan. ✉email: matsuoka@fbs.osaka-u.ac.jp; masahiroueda@fbs.osaka-u.ac.jp

The asymmetric shapes of migratory ameboid cells, which have an extending pseudopod and a contracting tail at the anterior and posterior, respectively, allow for efficient crawling movements on a substrate, thus providing a physiological basis for single-celled organisms to find nutrients, for neurons to migrate in the developing nervous system, for neutrophils to find and kill invading pathogens in the immune system, and other essential biological functions[1–4]. Extensive studies in various motile cells have revealed common features in the molecular components and their spatiotemporal dynamics for cell polarization, in which asymmetric signals are somehow self-organized in intracellular signaling pathways leading to cytoskeletal rearrangements for the polarized shape[5]. Activated Ras GTPases, phosphoinositide 3-kinases, and phosphatidylinositol lipid 3,4,5-trisphosphate $(PI(3,4,5)P_3)$ are localized at the leading edge of cells to regulate Rac-mediated actin polymerization for pseudopodia extension, while inactivated Ras GTPases, phosphatase and tensin homolog (PTEN), $PI(4,5)P_2$, and $PI(3,4)P_2$ are localized at the anterior to suppress lateral pseudopods[6,7]. These characteristic features in asymmetric signal generation for cell polarity and motility are shared among evolutionary distant organisms such as mammalian leukocytes and social amoebae *Dictyostelium discoideum*[5,8,9], although the underlying mechanisms of the asymmetric signaling remain unknown.

PTEN is a 3-phosphatase of $PI(3,4,5)P_3$ and is a key regulator of the $PI(3,4,5)P_3$ signaling pathway. PTEN is localized at the posterior membrane of a migrating cell, which is essential for $PI(3,4,5)P_3$-mediated signaling in various physiological phenomena, such as cell polarity and migration[10–13]. In *D. discoideum*, PTEN and $PI(3,4,5)P_3$ are localized in a mutually exclusive manner at the back and front regions of migrating cells, respectively[14,15]. Genetic disruption of PTEN enzymatic activity impairs the reciprocal polarity, leading to uniform covering of the whole-plasma membrane with $PI(3,4,5)P_3$, the loss of polarity, and multiple broad pseudopods, indicating the essential role of $PI(3,4,5)P_3$ degradation by PTEN in cell polarization[14]. Membrane binding by PTEN is suppressed at the anterior, but it is somehow stabilized at the posterior[16]. Besides the phosphatase domain, PTEN consists of an N-terminal $PI(4,5)P_2$-binding motif and C-terminal C2 domain, which contain stretches of basic residues[17,18]. These residues have been suggested to be responsible for electrostatic interactions with anionic phospholipids such as phosphatidylserine and $PI(4,5)P_2$ in PTEN recruitment to the membrane[19]. Because $PI(4,5)P_2$ is the enzymatic product of PTEN, the interaction between $PI(4,5)P_2$ and PTEN potentially promotes further PTEN recruitment to the membrane, leading to the formation of a positive feedback loop. In general, positive feedback regulation in a molecular reaction network responsible for cytoskeletal organization provides a mechanistic basis for cellular polarization[6]. However, it is difficult to characterize quantitatively the interaction between $PI(4,5)P_2$ and PTEN in living cells because of difficulty in artificially controlling the lipid composition. Consequently, whether the postulated positive feedback between PTEN and lipids exists physiologically remains unclear. To evaluate the electrostatic interactions between PTEN and $PI(4,5)P_2$, an assay system is required in which PTEN membrane binding can be characterized quantitatively with precise control of the lipid composition of the membrane.

We have developed single-molecule imaging techniques to analyze signaling mechanisms in living cells[20–23]. By using these techniques, we have characterized single-molecule behaviors of PTEN bound to the plasma membrane of living cells[16,24]. In *Dictyostelium* cells, PTEN adopts multiple binding states on the plasma membrane. PTEN has slower diffusion coefficients and longer lifetimes of membrane binding at the posterior of polarized cells than at the anterior, leading to its posterior accumulation.

Several basic amino acids of PTEN contribute to regulate the membrane-binding stability and mobility of PTEN on the plasma membrane, implying that electrostatic interactions with anionic lipids regulate the membrane-binding states of PTEN[19]. However, no direct evidence has demonstrated that anionic lipids affect the membrane-binding stability or intracellular mobility of PTEN.

Here we report an in vitro assay system for the single-molecule imaging analysis of PTEN on an artificial planer lipid bilayer, in which the membrane-binding stability and mobility of PTEN are characterized quantitatively under the precise control of the composition of anionic phospholipids, such as $PI(4,5)P_2$. We demonstrate that $PI(4,5)P_2$ extends the membrane-binding lifetimes and decreases the diffusion coefficients of PTEN in a basic residue-dependent manner. Furthermore, in vivo single-molecule imaging analysis of PTEN demonstrated that the lysine and arginine residues at the N-terminal domain of PTEN are essential for stabilizing the membrane binding of PTEN, cell polarity formation, and efficient migration of living cells. Our single-molecule imaging analysis of PTEN in vitro and in vivo consistently indicates the existence of positive feedback mechanism between PTEN and $PI(4,5)P_2$ for cellular polarization.

## Results

**Single-molecule imaging of PIP₂/PTEN in artificial membranes**. Artificial lipid bilayer membranes were formed on a hydrophilic glass surface and observed with total internal reflection fluorescence microscopy (TIRFM; Fig. 1a)[25,26]. To see whether the lipid bilayers are fluid and uniform for $PI(4,5)P_2$, we performed single-molecule diffusion measurements of fluorescently labeled $PI(4,5)P_2$. When the bilayer was doped with a fluorescent analog of $PI(4,5)P_2$ (TopFluor-$PI(4,5)P_2$, 450 ppb), which has its fluorophore in the fatty acid moiety, rapidly diffusing fluorescent spots were readily observed (Fig. 1b; Supplementary Movie 1). Trajectories of single TopFluor-$PI(4,5)P_2$ molecules were obtained by single-particle tracking (Fig. 1c). A probability density distribution of the displacement was well fitted to a single population distribution irrespective of the $PI(4,5)P_2$ concentration (Fig. 1d). Considering that fluorescent $PI(4,5)P_2$ could be incorporated into both leaflets of the artificial lipid bilayer, as observed in other lipid bilayer systems, the fitting indicates that the two leaflets exhibit identical fluidity with a single, homogeneous lipid phase under the tested $PI(4,5)P_2$-density conditions[27–29]. Consistent with this observation, the superimposition of the single-molecule trajectories revealed a vast region of the lipid bilayer with uniform $PI(4,5)P_2$ mobility (Fig. 1c). The lateral diffusion coefficient, $D$, of TopFluor-$PI(4,5)P_2$ was determined to be $5.2 \pm 0.6\ \mu m^2\ s^{-1}$ in 1 mol% $PI(4,5)P_2$ bilayers and $5.3 \pm 0.4\ \mu m^2\ s^{-1}$ in 20 mol% $PI(4,5)P_2$ bilayers, indicating that the $PI(4,5)P_2$ diffusion is independent of the $PI(4,5)P_2$ density (Fig. 1e). These values are consistent with those of lissamine rhodamine B-DOPE, which has its fluorophore in the polar head moiety (Supplementary Fig. 1) as well as $\beta$-BODIPY 530/550 HPC ($D = 8.5 \pm 4.9\ \mu m^2\ s^{-1}$)[30], and almost no aggregation of these lipid fluorescent analogs was detected in the fluorescence intensity distribution (Supplementary Fig. 2), showing that the site of fluorescent dye conjugation has negligible influence on the lipid mobility. Thus the bilayers used in this study were uniform and fluid, providing a reliable assay system for interactions between peripheral membrane proteins and lipids.

To observe single molecules of PTEN on the lipid bilayers, fluorescently labeled PTEN was prepared from *Dictyostelium* cells and added to the bilayers (Fig. 2a–c). After adding the labeled PTEN, a few bright fluorescent spots were observed on the 1 mol % $PI(4,5)P_2$ membrane (Fig. 2b), and a greater number of

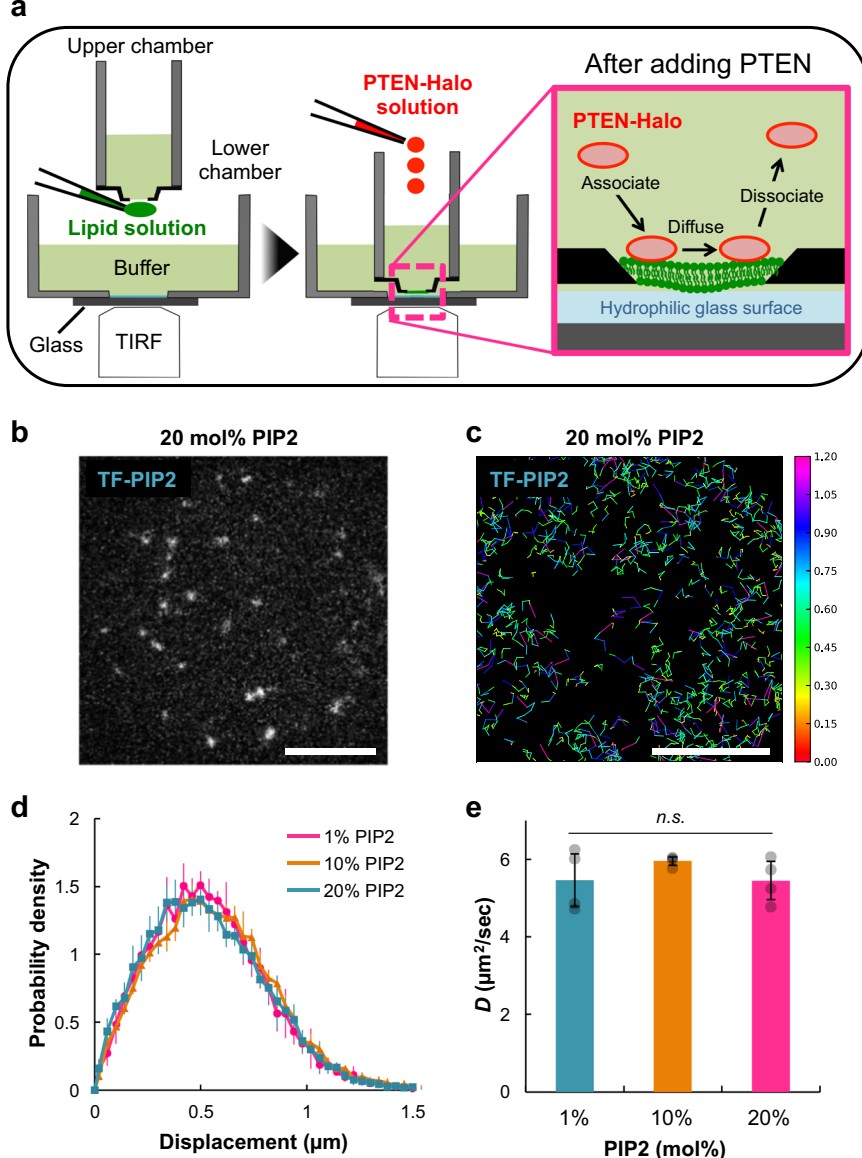

**Fig. 1 Lateral diffusion of PI(4,5)P$_2$ in artificial lipid bilayers. a** Schematic diagram describing the in vitro single-molecule imaging system using artificial planar lipid bilayers. **b** Representative TIRFM image of single TopFluor-PI(4,5)P$_2$ molecules in artificial lipid bilayers composed of 20 mol% PI(4,5)P$_2$. Scale bar, 10 μm. **c** Diffusion trajectories of single TopFluor-PI(4,5)P$_2$ molecules in lipid bilayers of 20 mol% PI(4,5)P$_2$ acquired at 16.5 ms per frame. Colors indicate the magnitude of the displacement within a unit time interval of $\Delta t = 16.5$ ms. Scale bar, 10 μm. **d** Probability density distribution of the displacement of single TopFluor-PI(4,5)P$_2$ molecules in lipid bilayers of 1 mol% PI(4,5)P$_2$ (magenta), 10 mol% PI(4,5)P$_2$ (orange), and 20 mol% PI(4,5)P$_2$ (blue). **e** Average diffusion coefficients quantified in **d**. $N = 2000$ (1 mol%), 2000 (10 mol%), 2000 molecules (20 mol%). $P = 0.46$ for 1 mol% PI(4,5)P$_2$ ($n = 4$ movies) versus 10 mol% PI(4,5)P$_2$ ($n = 4$ movies), $P = 1.00$ for 1 mol% PI(4,5)P$_2$ versus 20 mol% PI(4,5)P$_2$ ($n = 4$ movies), $P = 0.45$ for 10 mol% PI(4,5)P$_2$ versus 20 mol% PI(4,5)P$_2$ ($n = 4$ movies) by Tukey–Kramer test. Data are mean ± SD from 2 separate movie streams in 2 independent experiments ($n = 4$).

fluorescent spots were observed on the 10 mol% PI(4,5)P$_2$ membrane (Fig. 2c; Supplementary Movie 2). Most of the bright fluorescent spots had similar intensities and exhibited lateral diffusion freely on the PI(4,5)P$_2$-containing bilayer (Supplementary Figs. 1 and 2). This observation confirms that single bright fluorescent spots represent single molecules of labeled PTEN. Thus individual PTEN molecules can be observed in our assay system, in which the effects of PI(4,5)P$_2$ on PTEN membrane binding can be characterized quantitatively.

**PTEN membrane-binding parameters depend on PIP$_2$ density.** To examine how the interaction between PTEN and the membrane is regulated by PI(4,5)P$_2$, we quantified the diffusion

coefficient and membrane-binding lifetime on the lipid bilayer with different PI(4,5)P$_2$ densities. It is known that phosphoinositides constitute about 3 and 10 mol% of total lipids in the plasma membranes of mammalian cells and *Dictyostelium* cells, respectively[31,32], and that PI(4,5)P$_2$ is one of the major species of phosphoinositides on the plasma membrane. Thus, in this assay, we set the PI(4,5)P$_2$ density at three discrete values up to 10 mol%. At first, an average lateral diffusion coefficient of PTEN was estimated for each PI(4,5)P$_2$ density based on the probability density distribution of the displacement (Fig. 2d; see "Methods" for diffusion analysis). The diffusion coefficient was $5.0 \pm 0.4$ μm$^2$ s$^{-1}$ on the 1 mol% PI(4,5)P$_2$ membrane, which approximates that of a single TopFluor-PI(4,5)P$_2$ molecule ($D = 5.2 \pm 0.6$ μm$^2$ s$^{-1}$) (Fig. 1e). On the other hand, the diffusion coefficient was reduced

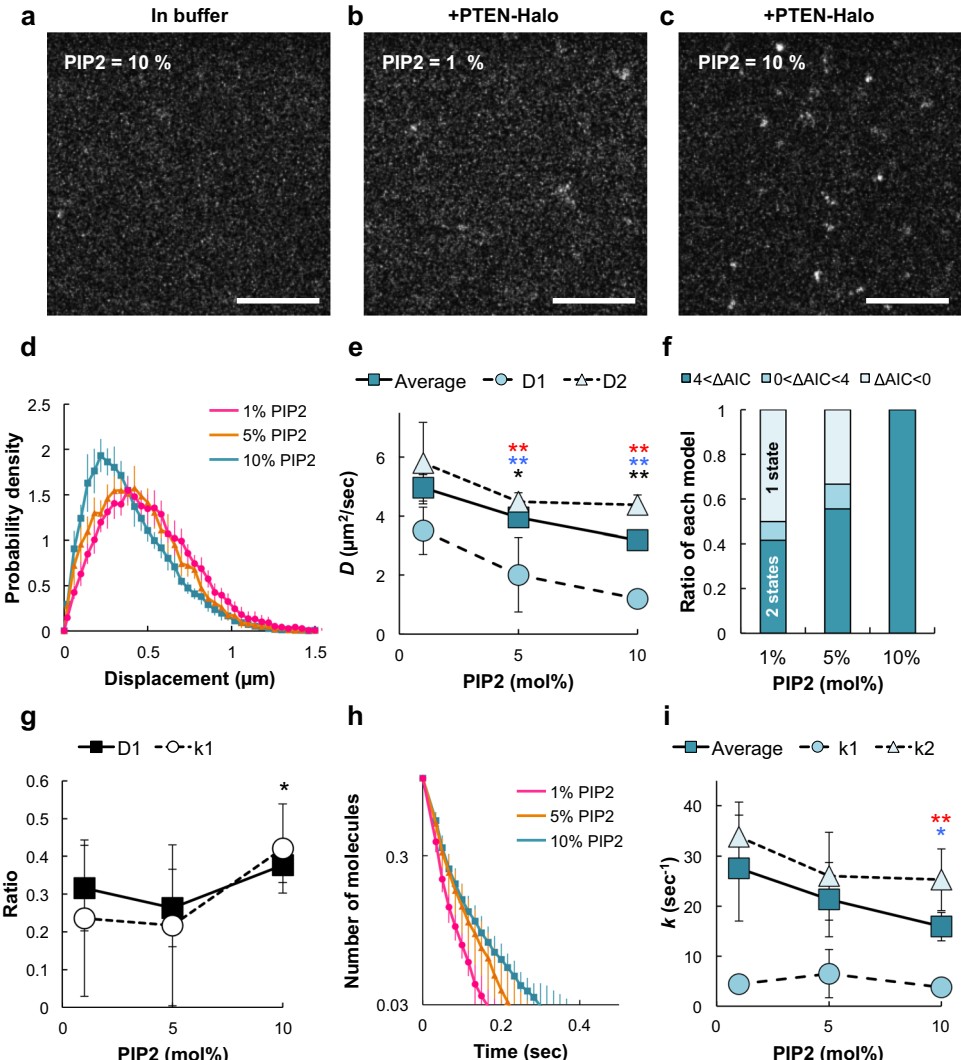

**Fig. 2 PI(4,5)P$_2$-induced stabilization of PTEN membrane binding. a–c** Representative TIRFM images of membranes composed of 10 mol% (**a**, **c**) or 1 mol% (**b**) PI(4,5)P$_2$ in the absence (**a**) or presence (**b**, **c**) of PTEN-Halo-TMR. Scale bars, 10 μm. **d** Probability density distribution of the displacement of single molecules in a unit time interval ($\Delta t = 16.5$ ms) on membrane containing 1 mol% (magenta), 5 mol% (orange), or 10 mol% (blue) PI(4,5)P$_2$. **e** PI(4,5)P$_2$ density dependency of the average diffusion coefficient (rectangle) and the diffusion coefficient for state 1 (circle) and state 2 (triangle). Welch's $t$ test; comparisons were made with "1 mol% PI(4,5)P$_2$" column for the same state. Black, blue, and red indicate $D_1$, $D_2$, and average $D$, respectively. **f** Estimation of the number of mobility states using the data obtained from each video. Light, middle, and dark colors indicate the ratio of the videos in which the AIC values show $AIC_1 - AIC_2 < 0$ (1 state model), $0 < AIC_1 - AIC_2 < 4$ (2 state model) and $4 < AIC_1 - AIC_2$ (2 state model). **g** PI(4,5)P$_2$ dependency of the number of molecules adopting the slower mobility state (rectangle) and the longer binding state (circle) normalized to the total number of molecules. Ratio of the slower mobility state ($D_1$, rectangle): $P = 0.31$ for 1 versus 5 mol% PI(4,5)P$_2$; $P = 0.13$ for 1 versus 10 mol% PI(4,5)P$_2$. Ratio of the longer binding state ($k_1$, circle): $P = 0.85$ for 1 versus 5 mol% PI(4,5)P$_2$; $P = 0.02$ for 1 versus 10 mol% PI(4,5)P$_2$. $P$ values were obtained by Welch's $t$ test. **h** Dissociation curve of single molecules on the membranes containing 1 mol% (magenta), 5 mol% (orange), or 10 mol% (blue) PI(4,5)P$_2$. **i** PI(4,5)P$_2$ dependency of the average dissociation rate constant (rectangle) and the dissociation rate constant for state 1 (circle) and state 2 (triangle). Welch's $t$ test; comparisons were made with "1 mol% PI(4,5)P$_2$" column for the same state. Blue and red indicate $D_2$ and average $D$, respectively. Data of 1, 5, and 10 mol% PI(4,5)P$_2$ are the mean ± SD from 12 (4 experiments, 6000 molecules), 9 (3 experiments, 45,000 molecules), and 9 (3 experiments, 45,000 molecules) separate movies, respectively.

to 3.2 ± 0.3 μm$^2$ s$^{-1}$ on the 10 mol% PI(4,5)P$_2$ membrane, showing that PTEN diffusion was about two times slower than PI(4,5)P$_2$ diffusion, which was approximately constant irrespective of the density (Fig. 2e, Average). Larger PI(4,5)P$_2$ densities increasingly retarded PTEN diffusion, reaching a plateau at 10 mol% PI(4,5)P$_2$.

We noticed that PTEN diffusion is better explained by assuming multiple diffusion coefficients, as we did previously in living *Dictyostelium* cells[16,19]. The displacement distribution was better fitted by a two-component probability density function (PDF) than a single-component function, as suggested by the Akaike Information Criterion (AIC) (Fig. 2d, f). The two

diffusion coefficients both decreased with increasing PI(4,5)P$_2$ density to 1.2 ± 0.3 ($D_1$) and 4.4 ± 0.3 μm$^2$ s$^{-1}$ ($D_2$), respectively, at 10 mol% and were smaller than the diffusion coefficient of PI(4,5)P$_2$ (Fig. 2e). The number of PTEN molecules exhibiting the slower mobility normalized to the total number of PTEN molecules was increased with increasing PI(4,5)P$_2$ density (Fig. 2g). Thus PI(4,5)P$_2$ slows down the lateral diffusion of individual PTEN molecules, and an increasing number of PI(4,5)P$_2$ molecules gradually affect the PTEN mobility.

Individual PTEN molecules only transiently associated with the lipid bilayers. We next examined the rate constant of PTEN membrane dissociation, which is an inverse of the membrane-

binding lifetime. An average rate constant was statistically estimated by measuring the duration of a single-molecule trajectory for all molecules observed under each PI(4,5)P$_2$ density (Fig. 2h). The rate constant was $27.6 \pm 10.6\,s^{-1}$ on the 1 mol% PI(4,5)P$_2$ membrane and $15.9 \pm 2.8\,s^{-1}$ on the 10 mol% PI(4,5)P$_2$ membrane, showing a prolongation of membrane-binding lifetime that depended on the PI(4,5)P$_2$ density (Fig. 2i, Average). The dissociation curves in Fig. 2h were characterized by multiple rate constants and fitted by a two-component exponential function[20,33]. The dissociation rate constant for the shorter binding state ($k_2$) was decreased with increasing PI(4,5)P$_2$ density from $33.8 \pm 6.9\,s^{-1}$ at 1 mol% to $25.3 \pm 6.2\,s^{-1}$ at 10 mol%, while that for the longer binding state ($k_1$) was not significantly changed with a minimum of $3.7 \pm 1.0\,s^{-1}$ (10 mol%) and the maximum of $6.5 \pm 4.8\,s^{-1}$ (5 mol%) (Fig. 2i). Like the number of PTEN molecules exhibiting slower mobility, the number of PTEN molecules exhibiting longer binding normalized to the total number of PTEN molecules was increased with increasing PI(4,5)P$_2$ density (Fig. 2g). Thus PI(4,5)P$_2$ stabilizes the membrane binding of individual PTEN molecules on the lipid bilayer. We speculated that PTEN becomes stably bound to the membrane as the number of PI(4,5)P$_2$ molecules interacting with PTEN increases[29,34]. Similar effects were observed for PI(4)P, a product of PTEN's catalytic activity for PI(3,4)P$_2$, but to a lesser extent than PI(4,5)P$_2$, because PI(4)P is less negatively charged than PI(4,5)P$_2$ (Supplementary Fig. 3).

**Electrostatic interactions stabilize PTEN membrane binding.** To verify that changes in the diffusion coefficient and membrane-binding lifetime were caused by direct interactions between PI(4,5)P$_2$ and PTEN, we examined single molecules of PTEN mutants with low PI(4,5)P$_2$-binding affinity, PTEN$_{R47A}$ and PTEN$_{N4}$, in which the basic residues of R47 and K11/K13/R14/R15, respectively, were substituted with alanine[18,35–37]. On the 1 mol% PI(4,5)P$_2$ membrane, lateral diffusion mobility was not significantly changed by the mutations, but with increasing PI(4,5)P$_2$ density the mobility of both mutants showed less sensitivity than the mobility of wild-type PTEN (Fig. 3a–c, j; Supplementary Movie 3). Of note, the average diffusion coefficient of PTEN$_{N4}$ showed negligible PI(4,5)P$_2$ dependency. On the 10 mol% PI(4,5)P$_2$ membrane, both mutants exhibited faster diffusion than wild type, as shown by the rightward shifts in the displacement distributions (Fig. 3a–d; Supplementary Movie 3). These shifts were mainly due to the increase in the diffusion coefficient for the slower mobility state, $D_1$, and the decrease in the normalized number of molecules exhibiting the slower mobility, which were more prominent in PTEN$_{N4}$ than PTEN$_{R47A}$ (Fig. 3e, f). This observation indicates that $D_1$ and $D_2$ are too similar to distinguish two distinct diffusion states in PTEN$_{R47A}$ and PTEN$_{N4}$. Consistently, AIC suggested two diffusion states for wild-type PTEN, but one diffusion state for each mutant (Fig. 3l). These results suggest that the lateral diffusion of PTEN is slowed down due to direct interactions with PI(4,5)P$_2$ via these basic residues. Notably, the four residues mutated in PTEN$_{N4}$ make a larger contribution to the PI(4,5)P$_2$ sensitivity than R47.

In addition, the membrane binding of both mutant PTENs was not stabilized to the same extent as wild-type PTEN with increasing PI(4,5)P$_2$ density (Fig. 3a–c, k). On the 10 mol% PI(4,5)P$_2$ membrane, both mutants exhibited faster dissociation than wild type, as shown by the leftward shifts of the dissociation curves (Fig. 3a–c, e). The faster dissociation of the mutants was due to the increase in the rate constants of both the longer and shorter binding states and the decrease in the normalized number of molecules exhibiting longer binding (Fig. 3h, i). Of note, the

longer binding state completely disappeared in PTEN$_{N4}$ (Fig. 3i), and the dissociation rate constant of PTEN$_{N4}$ showed negligible PI(4,5)P$_2$ dependency (Fig. 3k). These results suggest that the membrane binding of PTEN is stabilized through electrostatic interactions with PI(4,5)P$_2$. Therefore, the four residues in the N-terminal PI(4,5)P$_2$-binding motif may interact with more PI(4,5)P$_2$ molecules than does R47.

**PIP$_2$-binding residues enable stable membrane binding in vivo.** We next examined whether the lateral diffusion mobility and membrane-binding stability of PTEN are regulated by PI(4,5)P$_2$ on the plasma membrane of living cells similarly to the artificial lipid bilayer. Mutant or wild-type PTEN was expressed and labeled with tetramethylrhodamine (TMR) via HaloTag in living *D. discoideum* wild-type cells (Ax2) and observed on the basal plasma membrane with the same objective-type TIRFM as used in the in vitro assay. The three PTEN variants were compared under a constant condition, in which the PI(4,5)P$_2$ density was probed with GFP-Nodulin and kept almost constant due to the endogenous PTEN activity (Supplementary Fig. 4), and the PI(3,4,5)P$_3$ density was kept at basal level in less polarized cells due to the absence of pseudopod projections[38,39].

Single PTEN molecules were visualized as fluorescent spots when they were bound to the plasma membrane at the bottom of the cells (Fig. 4a–c; Supplementary Movie 4). The mutant PTENs exhibited faster diffusion than wild-type PTEN, as shown by the rightward shifts of the displacement distributions, which were fitted to a two-component PDF (Fig. 4d–g). The diffusion coefficients for the slower ($D_1$) and faster mobility ($D_2$) states were higher with the mutations (Fig. 4h). The normalized number of molecules exhibiting slower mobility was decreased more with PTEN$_{N4}$ than PTEN$_{R47A}$ (Fig. 4i). Thus the mutant PTENs exhibited faster diffusion mobility on average than wild-type PTEN, but the effect was more prominent in PTEN$_{N4}$ than in PTEN$_{R47A}$ (Fig. 4h, Average). These results indicate that the lateral diffusion of PTEN molecules is slowed owing to PI(4,5)P$_2$ binding on the plasma membrane of living cells just as it is on the artificial lipid bilayers, although the mobility on the plasma membrane was about tenfold smaller on average than on the artificial lipid bilayers. We attribute the difference to the viscosity of the membranes, which is roughly inversely correlated to the diffusion coefficient of the fluorescent lipid analogs: $5.2\,\mu m^2\,s^{-1}$ in the artificial membrane (Fig. 1e) and $0.78\,\mu m^2\,s^{-1}$ in the plasma membrane[40]. Under constant temperature, $D_1$ and $D_2$ of single PTEN molecules on the plasma membrane are estimated to be equivalent to those on the artificial membrane with 10 mol% PI(4,5)P$_2$ after correction for the effective viscosity (Supplementary Fig. 5).

The conservation between the two membrane systems was also seen in the membrane-binding lifetime (Fig. 4j–l). The mutant PTENs exhibited faster dissociation than wild-type PTEN, as shown by the leftward shifts of the dissociation curves (Fig. 4j). The rate constant for the shorter binding state ($k_2$) was increased by the mutations (Fig. 4k), whereas the fraction of molecules exhibiting longer binding was decreased and completely disappeared in PTEN$_{N4}$ (Fig. 4l). Thus the membrane dissociation on average descended in order of PTEN$_{N4}$, PTEN$_{R47A}$, and wild-type PTEN (Fig. 4k, Average). In addition, the membrane association frequency was also reduced by these mutations, which is consistent with previous studies (Fig. 4m)[16,24]. These results suggest that electrostatic interactions with PI(4,5)P$_2$ stabilize PTEN binding on the plasma membrane of living cells. The positive regulation of PTEN membrane binding by its enzymatic product, PI(4,5)P$_2$, implies a positive feedback mechanism for PI(4,5)P$_2$/PTEN accumulation on the membrane.

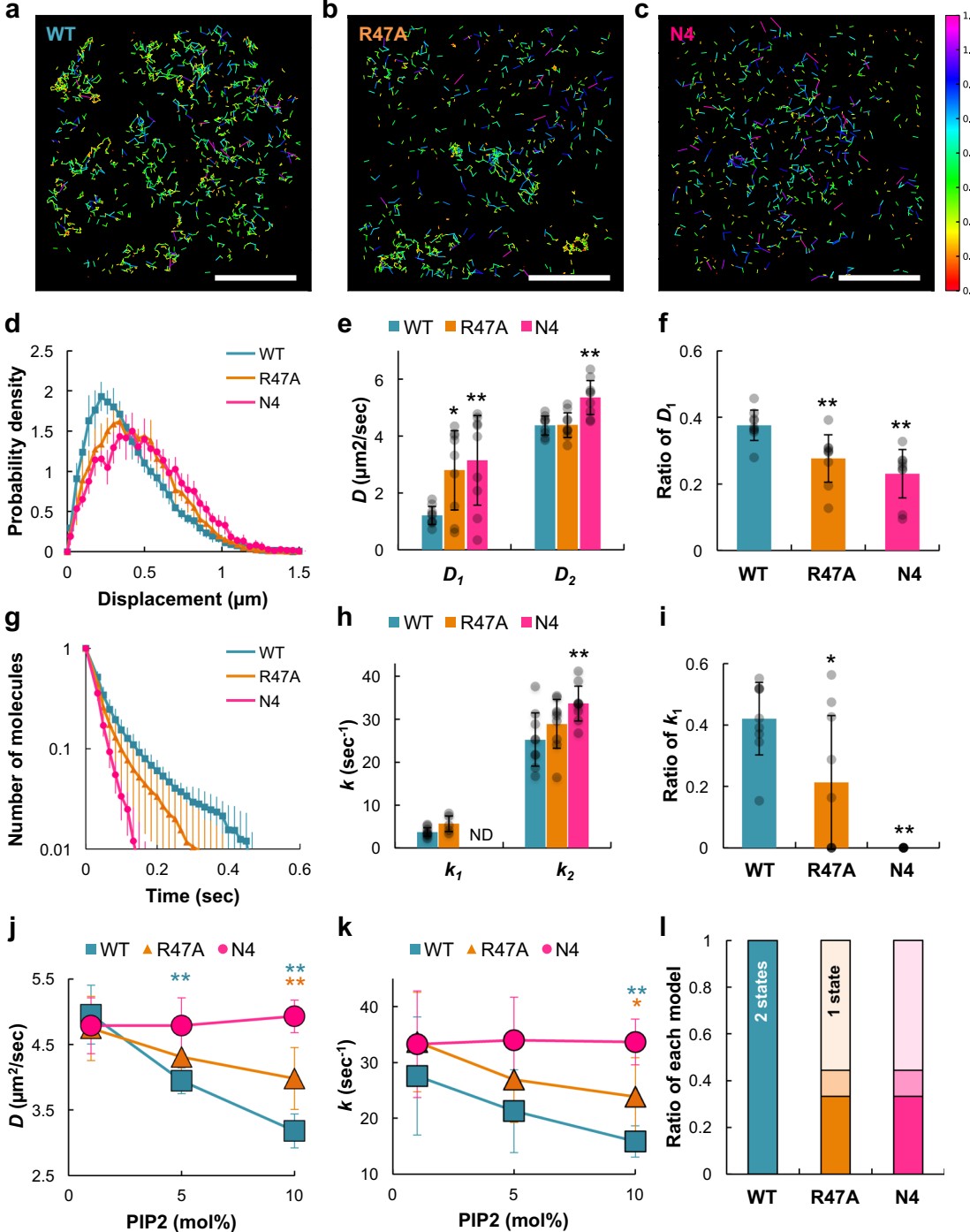

**Fig. 3 Membrane-binding stabilization via electrostatic interactions.** (**a–c**) Diffusion trajectories of single molecules of PTEN-Halo (**a**), PTEN$_{R47A}$-Halo (**b**), and PTEN$_{N4}$-Halo (**c**) labeled with TMR on artificial membranes of 10 mol% PI(4,5)P$_2$. Colors indicate the magnitude of the displacement of the molecule within a unit time interval of $\Delta t = 16.5$ ms. Scale bars, 10 μm. **d** Probability density distributions of the displacement ($\Delta t = 16.5$ ms) of single PTEN molecules on the membranes of 10 mol% PI(4,5)P$_2$. **e** Diffusion coefficients, $D_1$ and $D_2$, for slower and faster mobility states, respectively. Welch's $t$ test; comparisons were made with "WT" column for the same state. **f** The number of PTEN molecules adopting the slower mobility state normalized to the total number of molecules. Welch's $t$ test; comparisons were made with "WT" column. **g** Dissociation curves of single molecules on the membranes of 10 mol% PI(4,5)P$_2$. **h** Dissociation rate constants, $k_1$ and $k_2$, for the longer and shorter binding states, respectively. Welch's $t$ test; comparisons were made with "WT" column for the same state. **i** The number of PTEN molecules adopting the longer binding state normalized to the total number of molecules. Welch's $t$ test; comparisons were made with "WT" column. **j**, **k** PI(4,5)P$_2$ dependency of the average diffusion coefficient and average dissociation rate constant of PTEN (blue rectangle), PTEN$_{R47A}$ (orange triangle), and PTEN$_{N4}$ (magenta circle). Welch's $t$ test; comparisons were made with "1 mol% PI(4,5)P$_2$" column for the same cell types. Blue, orange, and magenta indicate PTEN$_{WT}$, PTEN$_{R47A}$, and PTEN$_{N4}$, respectively. **l** Estimation of the number of mobility states using the data obtained from each video (PI(4,5)P$_2$ density = 10 mol%). Light, middle, and dark colors indicate the ratio of the videos in which the AIC values show $AIC_1 - AIC_2 < 0$ (1 state model), $0 < AIC_1 - AIC_2 < 4$ (2 state model), and $4 < AIC_1 - AIC_2$ (2 state model). $N = 4500$ (1 mol%), 5997 (5 mol%), and 4365 (10 mol%) PTEN$_{R47A}$ molecules; 7056 (1 mol%), 4499 (5 mol%), and 4500 (10 mol%) PTEN$_{N4}$ molecules. Data are mean ± SD from 3 separate movies in 3 independent experiments ($n = 9$).

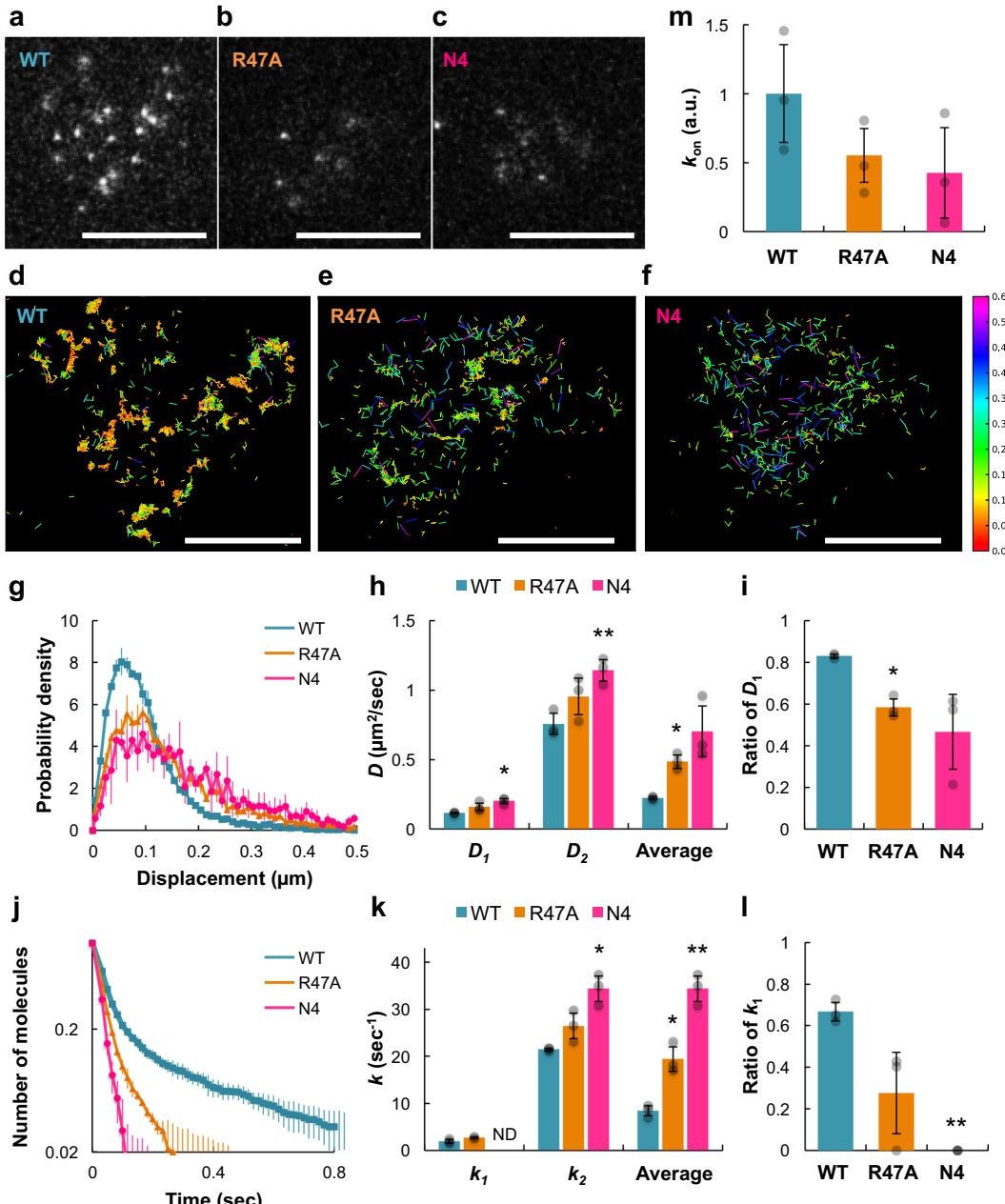

**Fig. 4 Membrane-binding stabilization on the plasma membrane of living *Dictyostelium* cells. a–c** Representative TIRFM images of *Dictyostelium discoideum* wild-type cells expressing PTEN-Halo (**a**), PTEN$_{R47A}$-Halo (**b**), and PTEN$_{N4}$-Halo (**c**) labeled with TMR. White spots represent single PTEN molecules bound to the plasma membrane at the bottom of the cell. Scale bars, 5 μm. **d–f** Diffusion trajectories of single molecules of PTEN (**d**), PTEN$_{R47A}$ (**e**), and PTEN$_{N4}$ (**f**) in single cells. Colors indicate the magnitude of displacement the molecule moved within a unit time interval of $\Delta t = 16.5$ ms. Scale bars, 5 μm. **g** Probability density distributions of the displacement ($\Delta t = 16.5$ ms) of single PTEN molecules on the plasma membranes. (**h**) Diffusion coefficients, $D_1$ and $D_2$, for the slower and faster mobility states, respectively. Welch's $t$ test; comparisons were made with "WT" column for the same state. **i** The number of PTEN molecules adopting the slower mobility state normalized to the total number of PTEN molecules. Welch's $t$ test; comparisons were made with "WT" column. **j** Dissociation curves of single PTEN molecules on the plasma membranes. (**k**) Dissociation rate constants, $k_1$ and $k_2$, for the longer and shorter binding states, respectively. Welch's $t$ test; comparisons were made with "WT" column for the same state. **l** The number of PTEN molecules adopting the longer binding state normalized to the total number of PTEN molecules. Welch's $t$ test; comparisons were made with "WT" column. **m** Relative association frequency of PTEN in wild-type cells. The number of PTEN molecules that appeared on the membrane during a unit time interval within a unit area was normalized by the fluorescence intensity of PTEN-Halo-TMR in the cytoplasm. In order to calculate the relative association frequency of stably bound molecules, the molecules with binding times <3 frames were excluded from the data analysis. Data are mean ± SD from 1500 molecules in 3 cells (PTEN), 1500 molecules in 3 cells (PTEN$_{R47A}$), and 1186 molecules in 3 cells (PTEN$_{N4}$).

**PIP$_2$-bound PTEN enhances PIP$_3$ polarity and cell motility.** We examined whether the stabilization of membrane binding affects the subcellular localization of PTEN in living *Dictyostelium* cells. Wild-type PTEN is localized to the plasma membrane in *Dictyostelium*, and the membrane localization is suppressed at the leading edge when *Dictyostelium* migrate[14]. At the border of the leading edge, the membrane localization of PTEN drastically increases toward the posterior in an all-or-none manner

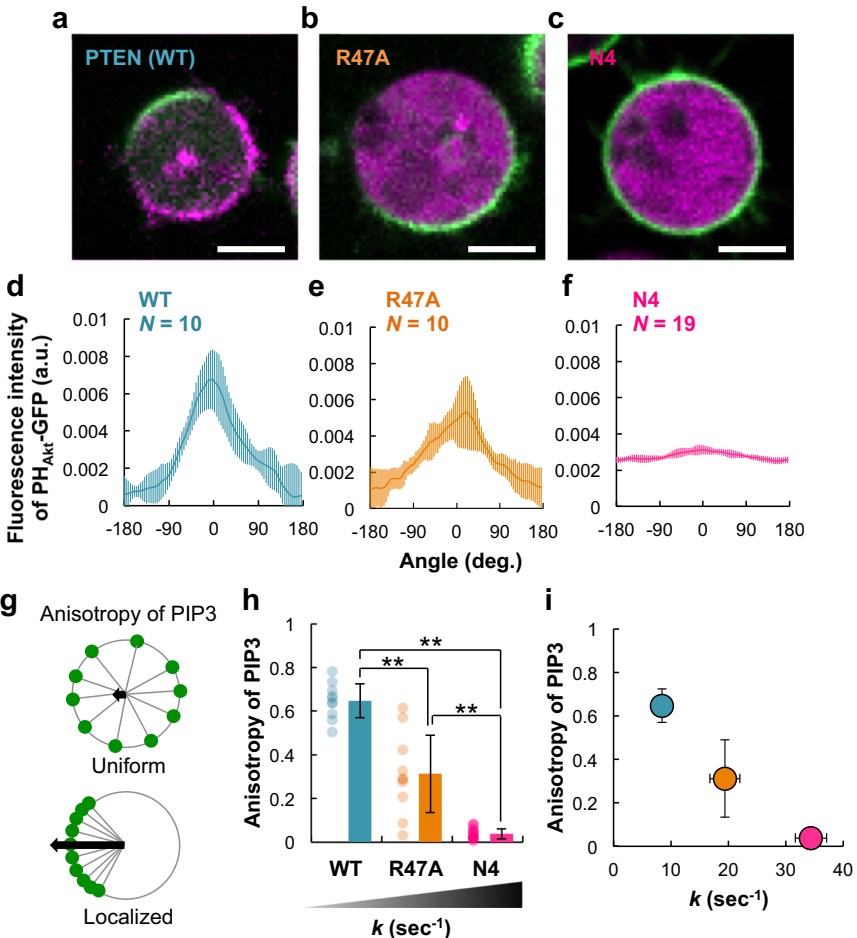

**Fig. 5 Confinement of the PI(3,4,5)P$_3$-enriched region through PI(4,5)P$_2$-induced accumulation of PTEN on the plasma membrane. a–c** Representative confocal microscopic images of *Dictyostelium discoideum pten*-null cells expressing PTEN-Halo (**a**), PTEN$_{R47A}$-Halo (**b**), and PTEN$_{N4}$-Halo (**c**) labeled with TMR (magenta) and PH$_{Akt/PKB}$-EGFP (green). The cells were treated with 5 μM latrunculin A and 4 mM caffeine. Scale bars, 5 μm. **d–f** Fluorescence intensity distributions of PH$_{Akt/PKB}$-EGFP on the plasma membrane of *pten*-null cells expressing PTEN (**d**, $n = 10$ cells), PTEN$_{R47A}$ (**e**, $n = 10$ cells), or PTEN$_{N4}$ (**f**, $n = 19$ cells) quantified along the cell periphery in **a–c**. **g** A schematic diagram describing the calculation of the anisotropy. **h** Anisotropy quantified in **b**. $P = 0.00$ for all combinations of cell types by Tukey–Kramer test. **i** Inverse linear correlation between the dissociation rate constant of PTEN and anisotropy of the PI(3,4,5)P$_3$ distribution. Data are mean ± SD.

(Supplementary Fig. 4), which can be explained by the bistability that arise due to the mutual inhibition between PTEN and PI(3,4,5)P$_3$ and a positive feedback loop amplifying PTEN membrane localization and PI(4,5)P$_2$ density[41,42]. PI(4,5)P$_2$ also shows an asymmetric distribution increasing toward the posterior, but the change is more gradual than PTEN. When PTEN$_{R47A}$ and PTEN$_{N4}$ were expressed in wild-type cells, they were mainly localized in the cytoplasm even if the membrane PI(4,5)P$_2$ density was unchanged (Supplementary Fig. 4). These results suggest that the stable interaction with PI(4,5)P$_2$ is critical for the amplification of PTEN membrane localization.

To examine how the failure in PTEN membrane accumulation affects the distribution of PI(3,4,5)P$_3$ and PI(4,5)P$_2$ on the plasma membrane, endogenous PTEN was replaced with either mutant PTEN by expressing the mutant in *pten*-null cells. We analyzed their spontaneous asymmetric localization in the absence of extracellular spatial cues using the fluorescently labeled Pleckstrin homology domain of Akt/PKB (PH$_{Akt/PKB}$-EGFP) and RFP-Nodulin, which are probes, respectively, specific for PI(3,4,5)P$_3$ and PI(4,5)P$_2$[43,44]. For convenience, the distribution was statistically analyzed in hemi-spherical cells that were adhered to the coverslip and did not show motile or protrusive activities. We obtained these cells by treatment with 5 μm of the actin

polymerization inhibitor latrunculin A, and the spontaneous asymmetric localization was induced by the addition of 4 mM caffeine, as reported previously[41,44]. PH$_{Akt/PKB}$-EGFP and PTEN-Halo-TMR exhibited asymmetric localization in a mutually exclusive manner in *pten*-null cells expressing wild-type PTEN (Fig. 5a). The PI(3,4,5)P$_3$ distribution, which was quantified by measuring the fluorescence intensity of PH$_{Akt/PKB}$-EGFP along the periphery of the cell image, showed a half-width of 95 degrees ($n = 10$ cells) (Fig. 5d). Anisotropy, which is an index of the localization and equals 0 or 1 when PI(3,4,5)P$_3$ exhibits a uniform or localized distribution, respectively, was $0.65 \pm 0.08$ (Fig. 5g,h). Under this condition, PH$_{Akt/PKB}$-EGFP and RFP-Nodulin showed anticorrelative localization so that the sum of the densities was conserved at every local point on the membrane (Supplementary Fig. 6)[42]. However, when endogenous PTEN was replaced with mutant PTEN, PI(3,4,5)P$_3$ became enriched on the membrane (Fig. 5b, c, e, f, h). In *pten*-null cells expressing PTEN$_{R47A}$, the PI(3,4,5)P$_3$ distribution was less localized than in cells expressing wild-type PTEN, showing a half-width of 166 degrees and anisotropy of $0.31 \pm 0.18$ ($n = 10$ cells) (Fig. 5b, e). PI(3,4,5)P$_3$ and PI(4,5)P$_2$ were anticorrelative, and the PI(4,5)P$_2$ distribution was reduced compared to that of PTEN expression, suggesting insufficient phosphatase activity on the membrane (Supplementary

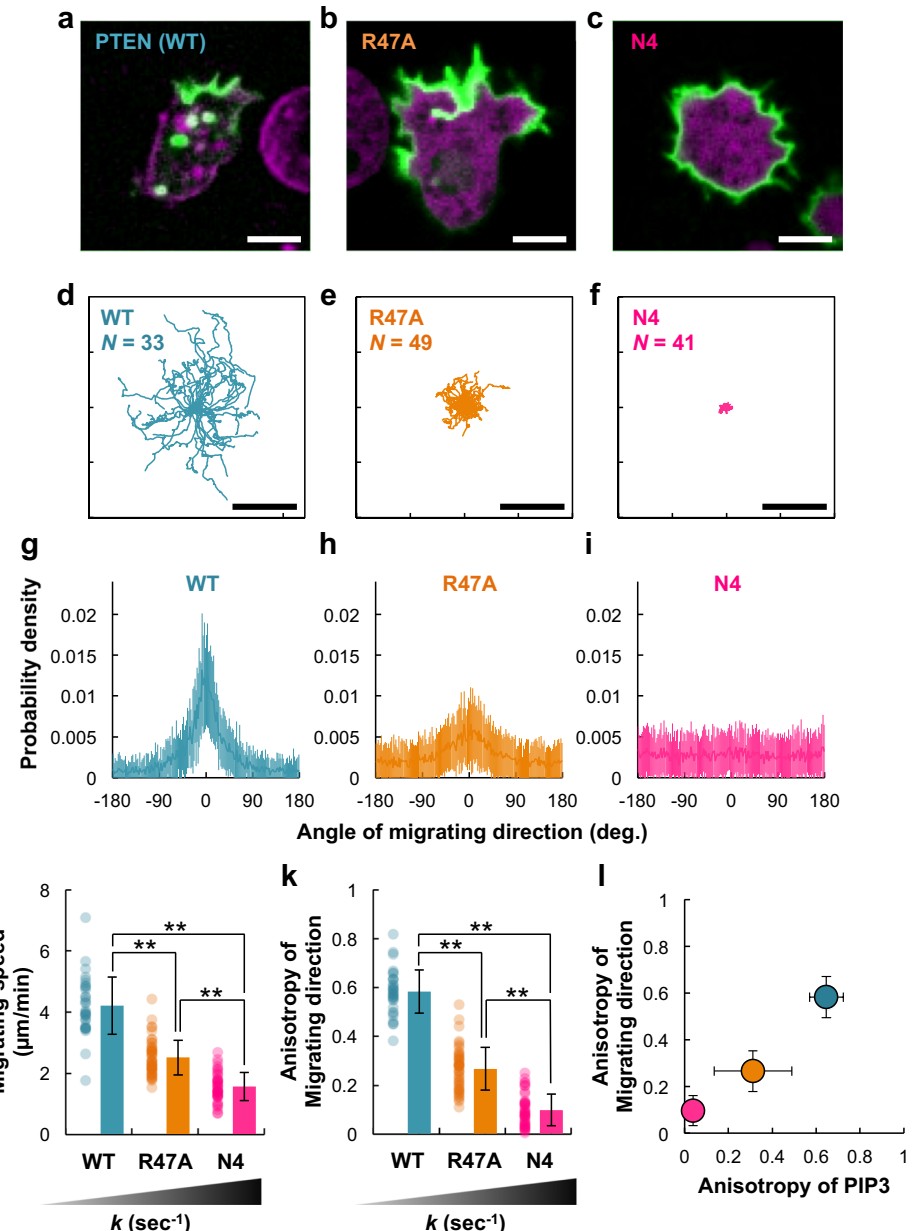

**Fig. 6 Efficient cell migration achieved through confinement of the PI(3,4,5)P₃-enriched region. a–c** Representative confocal microscopic images of *pten*-null cells expressing PTEN-Halo (**a**), PTEN$_{R47A}$-Halo (**b**), or PTEN$_{N4}$-Halo (**c**) labeled with TMR (magenta) and PH$_{Akt/PKB}$-EGFP (green) undergoing spontaneous migration. Scale bars, 5 μm. **d–f** Migration trajectories of *pten*-null cells expressing PTEN (**d**, $n = 33$ cells), PTEN$_{R47A}$ (**e**, $n = 49$ cells), or PTEN$_{N4}$ (**f**, $n = 41$ cells). The trajectories obtained at 5-s intervals for 15 min are shown. Scale bars, 30 μm. **g–i** Angle distributions of the migrating directions quantified in **d–f**, respectively. **j** Migration speed. $P = 0.00$ for all combinations of cell types by Tukey–Kramer test. **k** Anisotropy quantified in **g–i**. $P = 0.00$ for all combinations of cell types by Tukey–Kramer test. **l** Linear correlation between the anisotropy of the PI(3,4,5)P₃ distribution and the anisotropy of the migrating direction. Data are mean ± SD.

Fig. 6). In the case of PTEN$_{N4}$, PI(3,4,5)P₃ accumulated on almost the entire plasma membrane uniformly, and the anisotropy was $0.04 \pm 0.02$ ($n = 19$ cells) (Fig. 5c, f). We noticed that the anisotropy exhibited an inverse linear relationship with the rate constant for PTEN membrane dissociation (Fig. 5i). Phosphatase activities of the mutants were comparable to each other, although they were lower than that of wild-type, when assayed with soluble PI(3,4,5)P₃ in vitro (Supplementary Fig. 7). Therefore, the PI(3,4,5)P₃-enriched region is confined most likely by amplifying PTEN membrane localization through stabilization by PI(4,5)P₂.

Finally, we examined the spontaneous migration of *pten*-null cells in the absence of extracellular cues under the same observation

conditions as above without either latrunculin A or caffeine. Cells expressing wild-type PTEN exhibited a clear morphological anterior–posterior polarity with a single dominant pseudopod at the front (Fig. 6a). During migration, the angle of the migrating direction in wild-type PTEN-expressing cells was 0 degrees on average and had a relatively narrow distribution, suggesting that the cells tended to continue migrating in one direction for some time (Fig. 6d, g). The cells migrated effectively at an average migrating speed of $4.20\,\mu m\,min^{-1}$, and the anisotropy of the migrating direction was 0.58 (Fig. 6j, k). However, *pten*-null cells expressing mutant PTEN exhibited hyperactive pseudopod formation due to the unconfined PI(3,4,5)P₃ enrichment (Fig. 6b, c). In the case of

PTEN$_{R47A}$, the cells exhibited ambiguous anterior–posterior polarity with a broadened leading edge from which multiple pseudopods were projected. In the case of PTEN$_{N4}$, the cells hardly exhibited morphological polarity and isotropically projected multiple pseudopods. As a consequence, the cells could not migrate effectively (Fig. 6e, f, h, i). The average migrating speed and anisotropy of the migrating direction were 2.50 μm min$^{-1}$ and 0.27 for PTEN$_{R47A}$ and 1.55 μm min$^{-1}$ and 0.10 for PTEN$_{N4}$ (Fig. 6j, k). The two anisotropies of the PI(3,4,5)P$_3$ distribution and migrating direction showed a linear relationship, suggesting that confinement of the PI(3,4,5)P$_3$-enriched region correlates to the directedness of the cell migration (Fig. 6l). Thus the positive feedback loop accumulating PI(4,5)P$_2$ and PTEN on the plasma membrane essentially regulates the asymmetric PI(3,4,5)P$_3$ signaling in directed cell migration.

## Discussion

A growing number of studies have reported that PTEN is regulated by PI(4,5)P$_2$ and that PI(4,5)P$_2$ and PI(3,4,5)P$_3$ levels are maintained through positive feedback regulation in a wide spectrum of vital functions, including cell division and migration[45,46]. In addition, biochemical assays revealed sigmoidal kinetics of the dephosphorylation reactions against mono-dispersed PI(3,4,5)P$_3$[47] and PI(3,4,5)P$_3$ on solid-supported membranes[48], suggesting an enhancement of the catalysis by the product PI(4,5)P$_2$. Interaction with PI(4,5)P$_2$-containing vesicles causes structural changes of PTEN, as revealed by circular dichroism spectroscopy, further supporting this allosteric activation mechanism[49]. These processes are dependent on an N-terminal stretch of clustered basic residues, the so called PI(4,5)P$_2$-binding motif, suggesting a requirement for direct interactions with PI(4,5)P$_2$[50]. On the other hand, it is poorly understood how the interaction between PTEN and the lipid bilayer is regulated by PI(4,5)P$_2$. Several studies have revealed an essential role of the N-terminal motif in the accumulation of PTEN on PI(4,5)P$_2$-containing vesicles and the plasma membrane in living cells based on the steady-state ratio of the lipid bilayer-associated fraction[24,37]. Previously, we have shown quantitatively that the PI(4,5)P$_2$-binding motif is required for frequent membrane associations by single-molecule imaging analysis[16]. By using an in vitro single-molecule imaging assay with precise control of the lipid composition of the membrane, here we revealed that the membrane binding of PTEN is stabilized by direct interactions with PI(4,5)P$_2$, suggesting a positive feedback loop that amplifies the lipid bilayer-associated fraction (Figs. 2 and 3). Interactions between PTEN and PI(4,5)P$_2$ are conserved in vivo in eukaryotic cells, providing a self-organization mechanism for the spatial separation between PI(4,5)P$_2$ and PI(3,4,5)P$_3$ on the plasma membrane for cell polarization (Figs. 4–6). We have recently revealed that PTEN membrane binding becomes unstable due to the lipid phosphatase activity[41]. In fact, the average diffusion coefficient ($D = 0.20$ μm$^2$ s$^{-1}$) and the average dissociation rate constant ($k = 2.15$ s$^{-1}$) of PTEN$_{G129E}$, a catalytically dead mutant, were similar to those of wild-type PTEN ($D = 0.22$ μm$^2$ s$^{-1}$; $k = 1.94$ s$^{-1}$) in cells without stimulation, where the PI(3,4,5)P$_3$ level is minimized (Fig. 4)[16]. Therefore, we conclude that interactions with PI(4,5)P$_2$ and PI(3,4,5)P$_3$ have opposing positive and negative effects on the stability of PTEN membrane binding.

PTEN showed at least two diffusion coefficients on artificial lipid bilayers containing PI(4,5)P$_2$ (Fig. 2). Since PI(4,5)P$_2$ was distributed uniformly, exhibited a single diffusion coefficient (5.0 μm$^2$ s$^{-1}$), and did not make autonomous clusters under our observation conditions (Fig. 1), the variance in the PTEN diffusion coefficient most likely arose from the number of PI(4,5)P$_2$ molecules bound to a single PTEN molecule. It has been suggested that peripheral membrane proteins bound to a single lipid

molecule will have the same diffusion coefficient as the lipid molecule, indicating the number of PI(4,5)P$_2$ molecules bound to PTEN was 1 when PTEN adopts the faster mobility state on the 1 mol% PI(4,5)P$_2$ membrane ($D = 5.2$ μm$^2$ s$^{-1}$)[51]. The same study further reported that the lateral diffusion coefficient of PH domain oligomers, in which PH domains derived from GRP1 are tandemly connected with sufficiently long linkers, is roughly inversely proportional to the number of PI(3,4,5)P$_3$ molecules bound to the oligomer. The inverse relationship is based on the assumption of a string-of-beads structure of the PH domain oligomer, in which multiple lipid-binding sites are sufficiently separated and the friction acting via each site exerts additively when each site binds to the lipid on the membrane. If this assumption is applied to PTEN, then the number of bound PI(4,5)P$_2$ molecules will be estimated as 3 or 4 when PTEN adopts the slower mobility state on the 10 mol% PI(4,5)P$_2$ membrane ($D = 1.5$ μm$^2$ s$^{-1}$). Consistently, previous studies reported that R47 and K11/K13/R14/R15 possibly form two distinct sites for PI(4,5)P$_2$ binding, with K13 shared by both[18,35]. In addition, our results show that both R47A and N4 mutations cause a loss of the slower mobility state, possibly due to the reduction in the number of bound PI(4,5)P$_2$ molecules (Fig. 3), and that N4 leads to a more drastic change than R47A in the membrane-binding kinetics and phenotypes in vivo (Figs. 4–6). Since the diffusion coefficients of wild-type PTEN exhibit gradual changes upon P(4,5)P$_2$ density changes (Fig. 2), it is possible that PTEN induces PI(4,5)P$_2$ clustering and/or that PI(4,5)P$_2$ clusters recruit PTEN efficiently to the membrane.

The capability of interacting with multiple PI(4,5)P$_2$ molecules provides the possibility of mutual regulation between PTEN and PI(4,5)P$_2$ on the membrane. The lateral diffusion mobility of PTEN changes depending on the local density of PI(4,5)P$_2$. PTEN interacts with a smaller number of PI(4,5)P$_2$ molecules and moves faster under low PI(4,5)P$_2$ density conditions but interacts with a larger number of PI(4,5)P$_2$ molecules and moves slower under high PI(4,5)P$_2$ density conditions (Fig. 2). Thus individual PTEN molecules possibly exhibit temporal changes in their lateral diffusion mobility as the local PI(4,5)P$_2$ density changes on the membrane. In fact, such single-molecule behavior has been detected previously in the diffusion trajectories of PTEN and other proteins on both artificial lipid bilayers and the plasma membrane of living cells[16,34,51,52]. We found that membrane dissociation is suppressed when PTEN binds to multiple PI(4,5)P$_2$ molecules (Fig. 2e). Therefore, it is possible that the spatial heterogeneity in the local PI(4,5)P$_2$ density, which is frequently assumed in "raft" models in various cell types, causes a concentration of PTEN in the PI(4,5)P$_2$-enriched region. On the other hand, the spatial distribution of PI(4,5)P$_2$ also changes depending on the local density of PTEN. It is natural that the PI(4,5)P$_2$ density increases due to the enzymatic activity of PTEN. In addition, proteins capable of binding to multiple phospholipids simultaneously are known to induce lipid phase separation by clustering lipids on the bilayers[53–55]. Therefore, PTEN and PI-(4,5)P$_2$ likely form a positive feedback loop: PTEN generates heterogeneity in the PI(4,5)P$_2$ distribution on the membrane through local PI(4,5)P$_2$ production by its catalytic reaction and phase separation by its lateral diffusion, and PI(4,5)P$_2$ generates heterogeneity in the PTEN distribution by concentrating PTEN through the electrostatic interactions revealed by this study.

PTEN has been reported to be involved in epidermal growth factor signaling in several cancer cell lines by dephosphorylating PI(3,4)P$_2$[56]. Our measurement demonstrates that PTEN membrane binding is also stabilized by PI(4)P, a product of dephosphorylation against PI(3,4)P$_2$ as the substrate (Supplementary Fig. 3). Therefore, it is possible that multiple positive feedback loops can regulate PTEN activity in the presence of PI(4,5)P$_2$ and

$PI(3,4)P_2$. In *Dictyostelium* cells, $PI(3,4)P_2$ is another membrane marker of cellular anterior–posterior polarity[7]. $PI(3,4)P_2$ is localized at the posterior region of migrating cells. However, PTEN, when encoded by *ptenA* gene, is not essential for the metabolism of $PI(3,4)P_2$ in *Dictyostelium*, and there is no difference in the amount of $PI(3,4)P_2$ on the membrane between *pten*-null and wild-type cells. Therefore, the positive feedback loop involving $PI(4,5)P_2$ is likely to contribute dominantly to the cellular polarity of *Dictyostelium*, although other $PI(4)P$ molecules might work supportively.

Finally, the positive feedback mechanism of local PTEN and $PI(4,5)P_2$ accumulation could be a molecular basis for the self-organization of cellular polarization. Recently, we revealed mutual inhibitions between PTEN and $PI(3,4,5)P_3$ working in the spatial separation of two membrane regions where PTEN or $PI(3,4,5)P_3$ had accumulated[41]. We assumed that such spatial separation is promoted by the presence of positive feedback loops that enhance either the PTEN-enriched or $PI(3,4,5)P_3$-enriched states. Here we found direct interactions acting on the membrane to concentrate both $PI(4,5)P_2$ and PTEN at the posterior region of the cell. This observation does not necessarily exclude regulation from other proteins like MARCKS, which sequesters $PI(4,5)P_2$ in competition with PTEN at the leading edge[57]. The stronger the interaction between PTEN and $PI(4,5)P_2$, the more efficient the PTEN accumulation on the membrane (Figs. 3 and 4). The strength of the $PI(4,5)P_2$-PTEN positive feedback loop was experimentally manipulated by using mutant PTENs, which demonstrated that positive feedback regulates the size of the $PI(3,4,5)P_3$ domains and thereby the efficiency of cell migration (Figs. 5 and 6). Our comparative approach using both in vitro and in vivo assay systems to quantify the reaction kinetics and diffusion mobility at single molecule resolution provides a powerful tool to reveal the molecular mechanisms of the protein/lipid interactions, thus advancing understanding of the physiological roles of PTEN in cellular polarity formation.

## Methods

**Reagents**. Synthetic phospholipids 1,2-dioleoyl-*sn*-glycero-3-phosphocoline (18:1 DOPC, PC), 1,2-dioleoyl-*sn*-glycero-3-phospho-(1′-myo-inositol-4′,5′-bisphosphate) (18:1 $PI(4,5)P_2$) (ammonium salt), 1,2-dioleoyl-sn-glycero-3-phospho-(1′-myo-inositol-4′-phosphate) (18:1 $PI(4)P$) (ammonium salt), 1-oleoyl-2-{6-[4-(dipyrrometheneboron difluoride)butanoyl]amino} hexanoyl-*sn*-glycero-3-phosphoinositol-4,5-bisphosphate (TopFluor® $PI(4,5)P_2$) (ammonium salt), and 1,2-dioleoyl-sn-glycero-3-phosphoethanolamine-*N*-(lissamine rhodamine B sulfonyl) (LRB-DOPE) (ammonium salt) were obtained as powders from Avanti Polar Lipids (Alabaster, AL). $PI(4,5)P_2$ and TopFluor-$PI(4,5)P_2$ were dissolved in a 65:35:8 mixture of chloroform, methanol, and water. DOPC was dissolved in a 2:1 mixture of chloroform and methanol.

**Cell culture and constructs**. *D. discoideum* wild-type Ax2 was used as the parent strain. Cells were grown at 21 °C in HL5 medium (Formedium, UK) supplemented with 100 ng mL⁻¹ folic acid and 5 ng mL⁻¹ vitamin B12 in culture dishes kept still or flasks shaken at 175 rpm. Cells were transformed by electroporation as follows. Ten mL of cell culture at a cell density of $2.0–3.0 \times 10^6$ cells mL⁻¹ was collected in a centrifuge tube and centrifuged at $600 \times g$ for 1 min at 4 °C to precipitate the cells. After the supernatant was removed, the cells were suspended in 200 μL of E-buffer (20 mM Na/K2 phosphate buffer with 50 mM sucrose at pH 7.4) and centrifuged again. After the supernatant was removed, the cells were suspended in 600 μL of ice-cold E-buffer at a cell density of $2.5–5.0 \times 10^7$ cells mL⁻¹. Two hundred μL of the cell suspension was transferred to a new 1.5-mL Eppendorf tube, and 1–2 μg of the plasmid was added to the tube. The cell suspension and an electroporation cuvette (2 mm gap, NIPPON genetics) were cooled for 10 min on ice. After the cell suspension was transferred to the cuvette, the cells were electroporated by using ECM 830 Electroporator (BTX) at 500 V, 100 μs × 10 pulses (1-s intervals) and immediately cooled on ice for 5 min. Two μL of 100× healing solution (0.1 M CaCl₂, 0.1 M MgCl₂) was placed on a new 9-cm culture dish. The electroporated cell suspension was added to the dish, which was then incubated at room temperature for 10 min. Ten mL of HL5 medium was placed in the dish and cultured at 21 °C. Drugs for the selection of transformants were added 24 h after the electroporation. All manipulations were done on ice with the buffer, plasmid, and cuvette kept sufficiently cool. In order to select the transformed cells, the cells were cultured

at 21 °C for ≥1 week under G418 (10 mg mL⁻¹), Blasticidin S (10 mg mL⁻¹), or hygromycin (50 mg mL⁻¹)[58].

Expression vectors of mutant PTEN were constructed as follows. Primers used for engineering the constructs are shown in Supplementary Table 1. The DNA fragment (1 or 2) was PCR amplified by using the pairs of primer 1 and 4 or primer 2 and 3 and pHK12neo_PTEN-Halo-6×his as a template. The DNA fragment of mutant PTEN was PCR amplified by using the pair of primers 1 and 2 and mixed DNA fragment 1 and 2 as a template. The amplified mutant PTEN fragment was inserted into the *Xba*I and *Spe*I sites of pHK12neo by the In-Fusion method (Clontech Laboratories Inc.). The nucleotide sequence of all constructs was confirmed before transformation. All cell strains are available from the corresponding authors upon request.

**Protein purification**. *D. discoideum* PTEN was purified from *D. discoideum* cells overexpressing PTEN fused with Halo-tag® (Promega) and 6-histidine tag by using Ni-NTA affinity column chromatography (GE Healthcare), as described previously[17]. Briefly, approximately $10^8–10^9$ cells grown in shaking culture were collected, pelleted, and suspended in 3 mL of Lysis buffer (50 mM Tris-HCl pH 8.0, 500 mM NaCl, 5 mM β-melcaptoethanol, 10 mM Imidazole, protein inhibitor EDTA-free (Roche tablet)). After 30 μL of NP-40 (10%) was added to lyse the cell membrane, the cell suspension was incubated on ice for 30 min. The lysate was centrifuged at $15,000 \times g$ for 30 min at 4 °C to remove membrane fractions. The supernatant was collected and filtered through a 0.22-μm filter. The flow-through fraction was mixed with 200 μL of Ni-NTA beads washed with Lysis buffer and reacted for 1 h with rotation at 4 °C. The reaction mixture was centrifuged at $600 \times g$ for 3 min at 4 °C, and the supernatant was discarded. The beads were washed with 3 mL of Wash buffer (50 mM Tris-HCl pH 8.0, 300 mM NaCl, 20 mM Imidazole) and centrifuged at $600 \times g$ for 15 min at 4 °C. After the supernatant was discarded and 3 mL of Wash buffer was added, the suspension was mixed by inversion and centrifuged at $600 \times g$ for 4 min. After the supernatant was discarded, 3 mL of Wash buffer was added to the tube to resuspend the residual beads and transferred to the column. After the Wash buffer had flowed through the column, Elution buffer (50 mM Tris-HCl pH 8.0, 500 mM NaCl, 150 mM Imidazole) was added to the column in 200-μL increments, and PTEN-Halo proteins were eluted from the column. A final concentration of 2 mM dithiothreitol (DTT) was added to the eluted fractions. PTEN-Halo was labeled with 2 μm HaloTag ligands conjugated with TMR (Promega) for 15 min at 4 °C. The buffer was replaced with Storage buffer (50 mM Tris-HCl pH 8.0, 100 mM NaCl, 2 mM DTT) by using a Zeba Spin Desalting Column and aliquoted into low adsorption tubes. The samples were frozen in liquid nitrogen and stored at −80 °C. The mutant PTEN proteins were prepared similarly.

**Phosphatase assay**. Water-soluble diC8-$PI(3,4,5)P_3$ (Echelon) was used to measure PTEN phosphatase activity against $PI(3,4,5)P_3$. In all, 250 mM recombinant PTEN was reacted with 5 nmol diC8-$PI(3,4,5)P_3$ at 21 °C for 2 h with gentle shaking in 50 μL of PTEN storage buffer. After the reaction, 150 μL of BIOMOL® green reagent (ENZO) was added, and the mixture was shaken gently for 1 h. In order to quantify the phosphate released from the substrate by the dephosphorylation reaction, the absorbance at 620 nm was measured.

**Preparation of artificial planar lipid bilayer**. Artificial lipid bilayers were generated by using a laboratory-made device according to a previous report[25]. Briefly, the device was composed of two chambers: an upper chamber and a lower chamber. The upper chamber was made of a glass tube (diameter, 7 mm; height, approximately 10 mm). The upper end was open, and the bottom end was sealed with a thin polypropylene film (thickness, 0.3 mm; SEKISUI SEIKEI). To make a small hole with a diameter of approximately 100 μm, the top of the protrusion formed by pushing a stylus into the center of the bottom film was shaved with a razor. The position of the upper chamber was controlled by a micromanipulator (PCS-5000; Burleigh) so that it was located in the midst of the lower chamber placed on the microscope stage. The lower chamber was composed of a metal chamber (Attofluor Cell Chamber, A-7816; Molecular Probes) and a coverslip (diameter, 25 mm; MATSUNAMI). Before setting up the device, the coverslips were sonicated in 0.1 M KOH for 30 min, rinsed well with MilliQ, and washed twice in 100% EtOH by sonication for 30 min. After being washed thoroughly, the coverslips were coated with 0.2% hydrophilic polymer (lipidure-103; NOF CORPORATION) and then air-dried at room temperature. The upper chamber was filled with approximately 200 μL of reaction buffer (20 mM Tris-HCl pH 8.0, 300 mM NaCl, 20 mM Imidazole), and the lower chamber was filled with approximately 400 μL of reaction buffer. After painting a lipid solution (10 mg mL⁻¹ lipid per n-decane) on the hole, the upper chamber was lowered into the buffer filled in the lower chamber. When the upper and lower faces of the lipid solution covering the hole were in contact with the buffer, the lipid molecules were aligned, and a lipid bilayer was formed. The formed lipid bilayer region could be observed in the bright field. The upper chamber was then lowered until the lipid bilayer touched the glass surface. Adjusting the amount of solution in the upper and lower chambers changed the water pressure applied to the membrane, and the thickness and deflection of the lipid membrane could be controlled. Since the hydrostatic pressure conditions for stabilizing the membrane were slightly different every time, slight differences occurred in the

amount of solution in the chamber. If the membrane cracked during this operation, the upper chamber was immediately raised, and the above operation was repeated.

**Cell preparation for live imaging.** The cultured cells were washed twice with 1 mL of development buffer without $Ca^{2+}$ or $Mg^{2+}$ (DB−; 5 mM $Na_2HPO_4$ and 5 mM $KH_2PO_4$), plated at a density of $5 \times 10^6$ cells mL$^{-1}$ on a 35-mm dish (IWAKI) filled with 1 mL of DB (5 mM $Na_2HPO_4$, 5 mM $KH_2PO_4$, 2 mM $MgSO_4$, 0.2 mM $CaCl_2$), and incubated for 5–6 at 21 °C. During the last 30 min, 3 nM TMR-conjugated HaloTag ligand (Promega) was added to the cell suspension for observation under TIRFM and 2 μm ligand for confocal microscopy.

**Single-molecule imaging under TIRFM.** TIRF imaging was performed using an inverted microscope (Ti-E; Nikon) equipped with a laboratory-made objective-type TIRFM system[59], as described previously[42]. Laser sources for 488 and 561-nm excitation light were solid-state CW lasers (SAPPHIRE 488–20 and Compass 561–20, respectively; COHERENT). Lasers were guided to the back focal plane of the objective lens (CFI Apo TIRF ×60 Oil, N.A. 1.49; Nikon) through a back port of the inverted microscope. The images were captured by an EM-CCD camera (iXon3 897; Andor) after passing through a dual band laser split filter set (Di01-R488/ 561–25×36, Di02R561–25×36, FF01–525/45–25, and FF01–609/54–25; Semrock) and a ×4 intermediate magnification lens (VM Lens C-4×; Nikon) using iQ2 (Andor).

For in vitro experiments, after the bilayer formation the purified PTEN-Halo-TMR was added to the upper chamber. PTEN molecules were allowed to disperse well in the reaction buffer during a 5-min incubation. Images of $256 \times 256$ pixels were acquired at 16.5 ms per frame for 30 s. The power of the 488- and 561-nm lasers was set to ~2 and ~3 mW, respectively. All in vitro measurements were performed at room temperature (21 °C).

For in vivo experiments, the cells after vital staining were washed 5 times with 1 mL DB− and spread on a well-washed cover glass (diameter 25 mm, thickness 0.12–0.17 mm; Matsunami) fixed on a metal chamber (Attofluor Cell Chamber; Molecular Probes). Images of $256 \times 256$ pixels were acquired at 16.5 ms per frame for 30 s with the laser power kept at approximately 1.5 mW. All in vivo measurements were performed at room temperature.

**Subcellular localization imaging under confocal microscopy.** The spatio-temporal dynamics of fluorescent molecules bound to the plasma membrane were imaged using an inverted microscope (ECLIPSE Ti; Nikon) equipped with a confocal unit (CSU-W1; Yokogawa) and NIS-Elements AR (Nikon). Laser sources for 488 and 561-nm excitation light were solid CW lasers (OBIS 488 nm and OBIS 561 nm, respectively; COHERENT). Time-lapse imaging was performed using two sCMOS camera (ORCA-Flash4.0v3; Hamamatsu Photonics) and a ×60 oil immersion objective lens (Apo ×60 Oil, NA 1.40; Nikon). To visualize intracellular PTEN localization, Halo-tag® (Promega) fusion protein was vital-stained with 2 μm Halo-tag® TMR ligand for 30 min and washed 5 times with 1 mL DB− before observation. The cells were seeded on a 35-mm glass base dish (Grass 12φ, thickness 0.15–0.18 mm; IWAKI). In experiments that inhibited actin poly-merization, imaging was started after cell treatment with 4 mM caffeine and 5 μm latrunculin A (SIGMA) for 15 min. The exposure of each channel was set to 100 ms, and a confocalized image was obtained at 5-s intervals.

**Single-particle tracking.** The trajectories of single molecules were automatically obtained from single-molecule imaging movies by using the laboratory-made software[16,19]. Briefly, the two-dimensional intensity distribution of a single fluor-escent spot was fitted to a two-dimensional Gaussian distribution to determine the $x$, $y$-coordinates of the center, which was regarded as the position of the molecule. The position of all fluorescent spots in all frames in the movie was determined. Fluorescence signals with aberrant properties such as abnormally large sizes were automatically eliminated by the setting thresholds: those typically showing a dia-meter falling within the range of 298–1436 nm were kept for the analysis. When the distance between two fluorescent spots in adjacent frames was within typical values for diffusing molecules[60], they were regarded as the same fluorescent molecule. Molecules with a binding time of one frame were excluded from the data analysis.

**Single-molecule diffusion analysis.** The diffusion coefficients of single molecules were estimated as previously reported[16]. Briefly, the displacement, $r$, of the molecules was measured from the trajectories with the time window, $\Delta t$. The probability density distribution of the displacement was fitted to a PDF,

$$P(r, \Delta t) = \sum_{i=1}^{n} a_i \frac{r}{2D_i \Delta t + 2\varepsilon^2} \exp\left(-\frac{r^2}{4D_i \Delta t + 4\varepsilon^2}\right),$$

assuming $n$ different mobility states. $D_i$ and $a_i$ represent the diffusion coefficient of the $i$th mobility state and the number of displacements from $i$th state molecules normalized to the total number, respectively ($i = 1, 2, \ldots, n$). The average diffusion coefficient ($D$) was calculated by the following formula, $a_1 D_1 + a_2 D_2$. $\varepsilon$ represents the standard deviation of the measurement error and was obtained from the mean square displacement (MSD), MSD($\Delta t$) = $4D*\Delta t + 4\varepsilon^2$ [21].

In order to estimate the state number, $n$, AIC was used,

$$AIC = -2lnL + 2k,$$

where $L$ indicates the maximum likelihood estimation (MLE) and $k$ represents the number of parameters for each theoretical model[16].

**Single-molecule dissociation analysis.** The time length of a trajectory was measured for all fluorescent spots detected. The data were made into a cumulative distribution showing the number of molecules bound to the artificial lipid bilayers or the plasma membranes as a function of time after the appearance, which is designated as a dissociation curve. Dissociation rate constants were estimated by fitting the dissociation curve with a sum of exponential functions,

$$F(t) = \sum_{j=1}^{m} A_j \exp\left(-k_j t\right),$$

where $k_j$ represents the dissociation rate constant of the $j$th binding state and $A_j$ represents a weight applied upon the $j$th exponential function ($j = 1, 2, \ldots, m$)[20,33]. The inverse of $k_j$ is the lifetime of PTEN membrane binding, $\tau_j$. $A_j/k_j$ corresponds to the relative number of molecules adopting the $j$th binding state on average. Using the average dissociation rate, $k$, the inverse of the integral value of the dissociation curve, $(A_1/k_1 + A_2/k_2)^{-1}$, and the ratio of the $k_1$ state, $(A_1/k_1)(A_1/k_1 + A_2/k_2)^{-1}$, were calculated. All data were fitted with a two-component exponential function by using the least squares method. When $A_1$ was <0.001, the dissociation curves were fitted with a one-component exponential function for simplicity.

**Data analysis of PI(3,4,5)P$_3$ domain.** All images were processed and analyzed by Fiji and ImageJ. The fluorescence intensity of PH$_{Akt/PKB}$-EGFP along the circle contour was measured and plotted against the angle $\theta$, with the point showing the centroid of the fluorescence intensity distribution on the membrane defined as $\theta = 0$. The anisotropy of the PH$_{Akt/PKB}$-EGFP intensity distribution along the rounded cell contours was calculated from the following equation,

$$\text{Anisotropy} = \sqrt{\left(\sum_{i=0}^{N} R_i \cos\theta_i\right)^2 + \left(\sum_{i=0}^{N} R_i \sin\theta_i\right)^2},$$

where $\theta_i$ and $R_i$ indicate the $i$th angle and the normalized fluorescence intensity of PH$_{Akt/PKB}$-EGFP at angle $\theta_i$, respectively. Anisotropy ranged from 0 to 1, with an anisotropy approximating 0 indicating that PI(3,4,5)P$_3$ is uniformly distributed along the cell membrane. The localization of fluorescently labeled proteins on the plasma membrane was quantified as the membrane/cytoplasm ratio in Supple-mentary Fig. 4. The fluorescence intensity distribution along the periphery of the cell image was measured using our custom macro on ImageJ or Fiji, averaged per pixel, and divided by the averaged fluorescence intensity measured in the cytoplasm.

**Data analysis of cell migration assay.** Starved cells were suspended in DB at a cell density of $5.0 \times 10^4$ cells mL$^{-1}$, and 1 mL of the suspension was transferred to a 35-mm dish (IWAKI). After 10-min incubation to allow the cells to settle on the glass surface, the cells were observed under an inverted microscope (Olympus IX-71) equipped with phase-contrast optics. Time-lapse images were acquired through a ×20 objective lens (PHP 20X LCACHN; Nikon) at 5-s intervals for 15 min by using a digital camera (DIGITAL SIGHT DS-2MBW; Nikon) and NIS-Elements BR (Nikon).

The $x$, $y$-coordinates of the centroid of the cell were automatically obtained using the laboratory-made software previously described[44,61]. The migration speed was calculated from the positional change of the centroid with a unit time window of 5 s[62]. To analyze the straightness of cell migration, the angle of migration every 5 s was obtained from the cell migration trajectories. The angle of the migrating direction was calculated as the angle between a vector moving from the position of the cell at an arbitrary time point to the next position and a vector moving from the previous position to the present position. An angle approaching 0° indicated that the cell was moving straight ahead. Using the same methods as the PI(3,4,5)P$_3$ domain analysis, the anisotropy of the migrating angle was obtained from the normalized histogram of migrating angles.

**Statistics and reproducibility.** Values are expressed as the mean ± standard deviation containing a specified number of independent experiments, movies, or cells. Statistical significance of differences between groups was determined by Welch's $t$ test. Tukey–Kramer test was used for comparisons among three or more groups. Significance levels are *$P < 0.05$ and **$P < 0.01$. Details of the number of biological replicates and the assays are given in the figure legends.

**Reporting summary.** Further information on research design is available in the Nature Research Reporting Summary linked to this article.

## Data availability

The source data for the main figures are available in Supplementary Data 1–6. All other data supporting the findings of this study are available from the corresponding authors upon request.

## Code availability

The laboratory-made software, "G-Track," used to track single molecules and single cells was once sold from a venture company, G-angstrom Co., LTD., in Japan (http://www.seotools.jp/news/id_15419.html). However, the company has since closed, and the software is not commercially available. A custom Python code for analyzing the diffusion states from single-molecule trajectories using the EM algorithm and an ImageJ macro for drawing kymographs can be obtained from github (https://github.com/DaisukeYoshioka99/DataAnalysis). All other code is available from the corresponding authors upon reasonable request.

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

## Acknowledgements

GFP-Nodulin was kindly provided by Y. Miao and P. N. Devreotes (Johns Hopkins University School of Medicine, Baltimore). We thank T. Uyeda (National Institute of Advanced Industrial Science and Technology, Ibaraki, Japan) for the PHAKT/PKB-EGFP construct, and members of the Ueda laboratories for helpful suggestions. We thank P. Karagiannis for English editing (Sofia Science Writing, Japan). This research was supported by Japan Society for the Promotion of Science (JSPS) KAKENHI Grant JP25871120 (to S.M.), by Precursory Research for Embryonic Science and Technology (JST-PRESTO) from Japan Science and Technology Agency JPMJPR1879 (to S.M.), and by the Advanced Research and Development Programs for Medical Innovation (AMED-CREST) from Japan Agency for Medical Research and Development AMED JP18gm0910001 (to M.U.).

## Author contributions

D.Y. conducted the experiments, designed the experiments, and wrote the paper. S.F., H.K., and D.O. conducted the experiments. T.I. wrote the paper. S.M. and M.U. designed the experiments and wrote the paper.

## Competing interests

The authors declare no competing interests.
