## [Peer Review File · Communications Biology]

Editorial Note: *This manuscript has been previously reviewed at another Nature Research journal. This document only contains reviewer comments and rebuttal letters for versions considered at Communications Biology.*

REVIEWERS' COMMENTS:

Reviewer #1 (Remarks to the Author):

This is a very nice manuscript, and has been further improved by careful attention to my previous comments. I support rapid publication of this revised version.

Reviewer #2 (Remarks to the Author):

The article "PI(4,5)P2-mediated positive feedback accumulates PTEN on the cell membrane and reinforces directed cell migration" by Matsuoka and colleagues offers substantial proof to a longstanding question in the field of cell migration. The authors have designed a new in vitro single molecule imaging assay to elegantly answer this question. The paper is well-written and the conclusions are well-supported with sufficient data and sound statistical analyses. However, I would like to suggest the authors to consider the following points that might improve this article:

1) Pg. 3--"The asymmetric shapes of migratory amoeboid cells, which have an extending pseudopod and a contracting tail at the anterior and posterior, respectively, allow for efficient crawling movements on a substrate, thus providing a physiological basis for single-celled organisms to find nutrients, for neurons to migrate in the developing nervous system, for neutrophils to find and kill invading pathogens in the immune system, and other essential biological functions (Theveneau and Mayor 2012; Nourshargh and Alon 2014; Weninger, Biro, and Jain 2014)."

The above line is well explained and illustrated in a newly published review article by Li et al, Excitable networks controlling cell migration during development and disease [published online ahead of print, 2019 Dec 10]. *Semin Cell Dev Biol.* 2019;S1084-9521(19)30171-5. The authors

may cite this paper.

2) Pg. 3—"These characteristic features in asymmetric signal generation for cell polarity and motility are shared among evolutionary distant organisms such as mammalian leukocytes and social amoebae *Dictyostelium discoideum* (Artemenko, Lampert, and Devreotes 2014; Tang et al. 2014)...."

The authors may cite a recently published paper which highlights this point: Pal DS et al. The excitable signal transduction networks: movers and shapers of eukaryotic cell migration. *Int J Dev Biol.* 2019;63(8-9-10):407-416.

3) Pg. 4—"These residues have been suggested responsible for electrostatic interactions with anionic..." should be changed to "These residues have been suggested to be responsible for electrostatic interactions with anionic..."

4) Pg. 7—The authors have used fluorescent analog of PI(4,5)P₂ to test whether the artificial lipid bilayers are fluid and uniform for PI(4,5)P₂. How does the fluorescent analog of PI(4,5)P₂ compare to regular non-fluorescent PI(4,5)P₂ in terms of lateral diffusion? Are diffusion properties comparable between the two PI(4,5)P₂ types?

5) Pg. 7—Are the proportions of PI(4,5)P₂ in the artificial bilayers comparable to what is naturally found in *Dictyostelium* or mammalian membranes?

6) Pg. 8—After addition of labeled PTEN, the number of bright fluorescent spots increase on the 10 mol% PI(4,5)P₂ compared to 1 mol% (Fig. 2B-C). Do the PTEN fluorescent spots increase further in 20 mol% PI(4,5)P₂? Have the authors checked that?

7) Pg. 10—" Similar effects were observed for PI(4)P, a product of PTEN's catalytic activity for PI(3,4)P₂, but to a lesser extent than PI(4,5)P₂ (Fig. S3)." Could the authors speculate on why the effects are to a lesser extent for PI(4)P compared to PI(4,5)P₂? Maybe add a line here.

8) Pg. 11—The low PI(4,5)P₂-binding PTEN mutants displayed faster diffusion than wildtype PTEN. Would a catalytically inactive PTEN mutant have similar diffusion/membrane-binding

compared to wildtype PTEN? It might be an important control for the authors to do for this assay.

9) Pg. 24—"PTEN-Halo was labeled with 2 mM HaloTag ligands conjugated with tetramethylrhodamine (TMR; Promega) for 15 min at 4°C." Is it 2 mM or 2 μ M?

10) Pg. 47—Fig. 1D legend. Replace 1%, 10% or 20% with 1% PI(4,5)P₂, 10% PI(4,5)P₂ or 20% PI(4,5)P₂. The authors may make similar changes in other figures as well.

Reviewer #3 (Remarks to the Author):

The authors aimed to address whether PTEN accumulation at the rear of migrating cells is driven by a positive feedback loop between the phosphatase and its lipid product PI(4,5)P₂. This is a very important question because although it is known that PTEN binds PI(4,5)P₂ and this binding is important for its activity, experimental evidence for PTEN accumulation in the back of a polarized cell due to increasing production of PI(4,5)P₂ is lacking.

To address this question, the authors devised and carefully validated a very elegant system for tracking single PI(4,5)P₂ and PTEN molecules in vitro and demonstrated that increasing amounts of PI(4,5)P₂ lead to increased binding of PTEN to the membrane. Importantly, this is dependent on the previously established PI(4,5)P₂-binding region of PTEN. The authors were also able to demonstrate that PTEN binding to the membrane of living Dictyostelium cells is also mediated by the PI(4,5)P₂-binding region of the phosphatase. Not surprisingly, PTEN mutants that were unable to bind to the membrane, also failed to rescue aberrant polarization and migration of cells lacking PTEN.

The work makes a significant and valuable contribution to our understanding of how increased amounts of PI(4,5)P₂ can drive PTEN accumulation in an in vitro situation, where PI(4,5)P₂ levels are controlled and this is the only phosphoinositide present. Based on this, it does seem likely that PTEN would also similarly accumulate in living cells; however, current experiments do not definitively confirm this hypothesis. Although the in vitro evidence is very thorough and convincing, the study does not directly demonstrate the existence of a positive feedback loop

between PTEN and PI(4,5)P2 in living cells.

Here is a list of concerns and suggestions for the authors:

1. The authors addressed previous reviewers' comments, which improved the strength of their arguments. However, even with the added experiments, some of the conclusions remain overstated (for example, the title of the entire article or the last sentence on page 17: "The positive feedback is conserved in vivo..."). The in vivo data in this study does demonstrate that PTEN binding to the membrane depends on its ability to bind to PI(4,5)P2, but does not prove that increased PI(4,5)P2 produced by PTEN is responsible for recruiting more PTEN to the membrane. This point was brought up by reviewer 1, who suggested testing PTEN association with the membrane over time in the presence of PI(3,4,5)P3, which would be useful to show that PTEN can dephosphorylate the lipid to PI(4,5)P2, and further recruit more PTEN molecules. This experiment could be taken further by testing catalytically-dead PTEN (for example, C124S), which should show a constant amount of PTEN on the membrane over time due to lack of PI(4,5)P2 production. Unfortunately, this experiment was not feasible under the current experimental conditions. Thus, the question remains whether the positive feedback loop between PTEN and PI(4,5)P2 occurs in physiological systems. Since the current data cannot support this notion, the authors could consider toning down their conclusion of the existence of the positive feedback loop in vivo, even though this remains a very likely possibility given previous studies and the current work.

2. It would be helpful if the authors explained the choice of mutations to disrupt PI(4,5)P2 binding more explicitly. The N4 mutant is based on the paper by Wei et al. 2015 according to the authors' citations. However, this article used K13 mutant for their analysis. Why did the authors decide to use all four substitutions here? Is the N4 mutant more deficient in PI(4,5)P2 binding than K13 mutant alone? This could be clarified in the text.

3. The authors attempted a valuable experiment of targeting mutant PTEN that is not able to interact with PI(4,5)P2 to the membrane to confirm that these mutants cannot rescue pten-null phenotype only because they cannot be recruited to PI(4,5)P2. Unfortunately, this experiment did not work. While this is understandable, the explanation in the paper is not clear. In the reviewer comments, the authors clearly stated that myristoylated wild-type PTEN is not

functional. This means that this technique is not suitable for recruitment of PTEN to the membrane and thus should not be used as support for the statement that "...the mutants could not rescue the pten-null phenotype even after membrane targeting...". Unless the authors find a different way to recruit PTEN to the membrane while retaining its catalytic activity, membrane targeting is not clarifying anything in this case. What does help the authors, and could be further emphasized, is that since the phosphatase activity of the two mutants is similar, yet they produce different phenotypes, reduced phosphatase activity alone cannot explain behavior of both mutants. Thus, it is likely that the difference is due to the relative defect in PI(4,5)P₂ binding (more severe for N4, less severe for K13 mutant), which strengthens the authors' conclusions.

4. The method for quantifying membrane/cytoplasm ratio of PTEN and Nodulin shown in Figure S4 was not described in sufficient detail.

Response to the reviewers' comments:

Our response to Reviewer #2:

We are grateful to reviewer #2 for the insightful comments that have helped improve our paper. We write the comments made by the reviewer in *bold italics* with our responses below.

1) Pg. 3--“The asymmetric shapes of migratory amoeboid cells, which have an extending pseudopod and a contracting tail at the anterior and posterior, respectively, allow for efficient crawling movements on a substrate, thus providing a physiological basis for single-celled organisms to find nutrients, for neurons to migrate in the developing nervous system, for neutrophils to find and kill invading pathogens in the immune system, and other essential biological functions (Theveneau and Mayor 2012; Nourshargh and Alon 2014; Weninger, Biro, and Jain 2014).”

The above line is well explained and illustrated in a newly published review article by Li et al, Excitable networks controlling cell migration during development and disease [published online ahead of print, 2019 Dec 10]. Semin Cell Dev Biol. 2019;S1084-9521(19)30171-5. The authors may cite this paper.

We thank the reviewer for recommending the newly published article. We have referenced the article in our revised manuscript (line 5, page 3).

2) Pg. 3—“These characteristic features in asymmetric signal generation for cell polarity and motility are shared among evolutionary distant organisms such as mammalian leukocytes and social amoebae Dictyostelium discoideum (Artemenko, Lampert, and Devreotes 2014; Tang et al. 2014)....”

The authors may cite a recently published paper which highlights this point: Pal DS et

al. The excitable signal transduction networks: movers and shapers of eukaryotic cell migration. Int J Dev Biol. 2019;63(8-9-10):407–416.

We thank the reviewer for the recommendation. We have referenced the article in our revised manuscript (line 15, page 3).

3) Pg. 4—"These residues have been suggested responsible for electrostatic interactions with anionic..." should be changed to "These residues have been suggested to be responsible for electrostatic interactions with anionic..."

We thank the reviewer for the suggestion. We have rewritten the sentence in our revised manuscript. (line 8, page 4).

4) Pg. 7—The authors have used fluorescent analog of PI(4,5)P₂ to test whether the artificial lipid bilayers are fluid and uniform for PI(4,5)P₂. How does the fluorescent analog of PI(4,5)P₂ compare to regular non-fluorescent PI(4,5)P₂ in terms of lateral diffusion? Are diffusion properties comparable between the two PI(4,5)P₂ types?

We appreciate the reviewer's concern. Indeed, it is possible that steric hindrance caused by the fluorophore affects the lateral diffusion properties of PI(4,5)P₂. Since we cannot quantify the diffusion of non-fluorescent PI(4,5)P₂, we compared two fluorescent analogs, TopFluor-PI(4,5)P₂, which has the fluorophore in the fatty acid moiety, and 18:1 LRB-DOPE, which has the fluorophore in the polar head moiety. We found that both analogs showed comparable diffusion coefficients and did not change their diffusion coefficients upon concentration changes (**Supplementary Figure 1**). Furthermore, despite the fact that the lateral diffusion and phase separation of lipid molecules are highly sensitive to the structure of the fatty acid moiety, we could not find differences between the two analogs. Therefore, we consider any influence by the fluorophore conjugation on the lateral diffusion of PI(4,5)P₂ to be negligible.

5) Pg. 7—Are the proportions of PI(4,5)P₂ in the artificial bilayers comparable to what is naturally found in Dictyostelium or mammalian membranes?

We thank the reviewer for the question. It is known that phosphoinositides constitute about 3% and 10% of total lipids in the plasma membranes of mammalian cells and *Dictyostelium* cells, respectively (van Meer *et al.*, *Nat. Rev. Mol. Cell Biol.*, 2008; Weeks *et al.*, *J. Lipid Res.*, 1980), and that PI(4,5)P₂ is one of the major species of phosphoinositides on the plasma membrane. Therefore, we believe that our experimental setup covers the physiological range of PI(4,5)P₂ density. We have included an explanation in the revised manuscript (line 17, page 7).

6) Pg. 8—After addition of labeled PTEN, the number of bright fluorescent spots increase on the 10 mol% PI(4,5)P₂ compared to 1 mol% (Fig. 2B-C). Do the PTEN fluorescent spots increase further in 20 mol% PI(4,5)P₂? Have the authors checked that?

We performed single-molecule imaging of PTEN on 20% PI(4,5)P₂ membrane and confirmed that the density, diffusion coefficient and dissociation rate constant of membrane-bound PTEN molecules plateaued at 10% PI(4,5)P₂. We noticed that the 20% PI(4,5)P₂ membrane was structurally unstable, with a PI(4,5)P₂-aggregated region sometimes appearing. Therefore, we decided not to include the data in the manuscript.

7) Pg. 10—" Similar effects were observed for PI(4)P, a product of PTEN's catalytic activity for PI(3,4)P₂, but to a lesser extent than PI(4,5)P₂ (Fig. S3)." Could the authors speculate on why the effects are to a lesser extent for PI(4)P compared to PI(4,5)P₂? Maybe add a line here.

We thank the reviewer for the comment. We speculate that the electrostatic interaction between PTEN and PI(4)P is weaker than that between PTEN and PI(4,5)P₂, because

PI(4)P is less negatively charged than PI(4,5)P₂. We have included an explanation in the revised manuscript (line 12, page 9).

8) Pg. 11—The low PI(4,5)P₂-binding PTEN mutants displayed faster diffusion than wildtype PTEN. Would a catalytically inactive PTEN mutant have similar diffusion/membrane-binding compared to wildtype PTEN? It might be an important control for the authors to do for this assay.

We thank the reviewer for the insightful comment. Previously, we measured the diffusion and membrane-dissociation kinetics of mutant PTEN in *Dictyostelium* cells, whose substrate binding and phosphatase activity were lost by G129E substitution (Matsuoka S. *et al.*, *PLoS Comput Biol.*, 2013). The average diffusion coefficient ($D = 0.20 \mu\text{m}^2 \text{sec}^{-1}$) and the average dissociation rate constant ($k = 2.15 \text{sec}^{-1}$) of PTEN_{G129E} were similar to those of wild-type PTEN ($D = 0.22 \mu\text{m}^2 \text{sec}^{-1}$; $k = 1.94 \text{sec}^{-1}$) measured in this study (**Fig. 4**). These measurements were performed in the cells without stimulation, where the PI(3,4,5)P₃ levels were minimized. However, we have reported that the membrane-binding becomes unstable in the presence of PI(3,4,5)P₃ dephosphorylation (Matsuoka and Ueda, *Nat. Commun.*, 2018). Therefore, we concluded that interactions with PI(4,5)P₂ and PI(3,4,5)P₃ have opposite effects, positive and negative, respectively, on the stability of PTEN membrane-binding. We have included the above discussion in the revised manuscript (line 22, page 16).

9) Pg. 24—"PTEN-Halo was labeled with 2 mM HaloTag ligands conjugated with tetramethylrhodamine (TMR; Promega) for 15 min at 4°C." Is it 2 mM or 2 μM?

We have corrected the typo (line 19, page 23).

10) Pg. 47—Fig. 1D legend. Replace 1%, 10% or 20% with 1% PI(4,5)P₂, 10% PI(4,5)P₂ or 20% PI(4,5)P₂. The authors may make similar changes in other figures

as well.

We have corrected the legends of all figures (Fig. 1-2, Supplementary Figure 1-2).

Our response to Reviewer #3:

We are grateful to reviewer #3 for the critical and informative comments that have helped improve our manuscript. As indicated in the responses that follow, we have addressed the comments in the revised version of our paper. We put the comments made by the reviewer in ***bold italics*** with our responses below.

(1) The authors addressed previous reviewers' comments, which improved the strength of their arguments. However, even with the added experiments, some of the conclusions remain overstated (for example, the title of the entire article or the last sentence on page 17: "The positive feedback is conserved in vivo..."). The in vivo data in this study does demonstrate that PTEN binding to the membrane depends on its ability to bind to PI(4,5)P₂, but does not prove that increased PI(4,5)P₂ produced by PTEN is responsible for recruiting more PTEN to the membrane. This point was brought up by reviewer 1, who suggested testing PTEN association with the membrane over time in the presence of PI(3,4,5)P₃, which would be useful to show that PTEN can dephosphorylate the lipid to PI(4,5)P₂, and further recruit more PTEN molecules. This experiment could be taken further by testing catalytically-dead PTEN (for example, C124S), which should show a constant amount of PTEN on the membrane over time due to lack of PI(4,5)P₂ production. Unfortunately, this experiment was not feasible under the current experimental conditions. Thus, the question remains whether the positive feedback loop between PTEN and PI(4,5)P₂ occurs in physiological systems. Since the current data cannot support this notion, the authors could consider toning down their conclusion of the existence of the positive feedback loop in vivo, even though this remains a very likely possibility given previous studies and the current work.

We thank the reviewer for the comment. We have rewritten the sentences to tone down the conclusion about the positive feedback between PTEN and PI(4,5)P₂ *in vivo* (line 19, page 16).

(2) It would be helpful if the authors explained the choice of mutations to disrupt PI(4,5)P₂ binding more explicitly. The N4 mutant is based on the paper by Wei et al. 2015 according to the authors' citations. However, this article used K13 mutant for their analysis. Why did the authors decide to use all four substitutions here? Is the N4 mutant more deficient in PI(4,5)P₂ binding than K13 mutant alone? This could be clarified in the text.

We thank the reviewer for the comment. We chose the N4 mutant according to the literature (Walker et al. 2004, Wei et al. 2015, as well as Das et al. 2003, which was missing from the reference list in the original manuscript). In our experiments, the phenotype of *pten*-null *Dictyostelium* including slow growth, aberrant cell morphology and loss of aggregation was almost recovered by the expression of PTEN_{K13A}-Halo, but not by PTEN_{N4}-Halo. Therefore, we considered the K13A mutant to not be significantly defective in PI(4,5)P₂ binding, and thus we decided to focus on the N4 mutant. The differences between the K13A and N4 mutant especially in PI(4,5)P₂ binding have not been investigated to date. In order to avoid confusion, we cite Das et al. 2003 and Walker et al. 2004 when the mutants are introduced in the revised manuscript (line 19, page 9).

*3) The authors attempted a valuable experiment of targeting mutant PTEN that is not able to interact with PI(4,5)P₂ to the membrane to confirm that these mutants cannot rescue *pten*-null phenotype only because they cannot be recruited to PI(4,5)P₂. Unfortunately, this experiment did not work. While this is understandable, the explanation in the paper is not clear. In the reviewer comments, the authors clearly stated that myristoylated wild-type PTEN is not functional. This means that this technique is not suitable for recruitment of PTEN to the membrane and thus should not be used as support for the statement that "...the mutants could not rescue the *pten*-null phenotype even after membrane targeting...". Unless the authors find a different way to recruit PTEN to the membrane while retaining its catalytic activity, membrane targeting is not clarifying anything in this case. What does help the authors, and could be further emphasized, is that since the phosphatase activity of the two*

mutants is similar, yet they produce different phenotypes, reduced phosphatase activity alone cannot explain behavior of both mutants. Thus, it is likely that the difference is due to the relative defect in PI(4,5)P2 binding (more severe for N4, less severe for K13 mutant), which strengthens the authors' conclusions.

We appreciate the reviewer's concern about the experimental data. According to the reviewer's suggestion, we have excluded from the manuscript the description about the enzyme activity assay using myristoylated PTEN in live cells.

(4) The method for quantifying membrane/cytoplasm ratio of PTEN and Nodulin shown in Figure S4 was not described in sufficient detail.

We thank the reviewer for the comment. We have included the method for quantifying the membrane/cytoplasm ratio of fluorescently labeled proteins in the revised manuscript (line 7, page 30).